# Novel Use of Social Media Big Data and Artificial Intelligence for Community Resilience Assessment (CRA) in University Towns

Mohammed Abdul-Rahman [1,2,3,*], Mayowa I. Adegoriola [1], Wilson Kodwo McWilson [1], Oluwole Soyinka [4] and Yusuf A. Adenle [5]

1 Department of Building and Real Estate, The Hong Kong Polytechnic University, Hong Kong, China
2 Department of Urban and Regional Planning, University of Lagos, Lagos 101017, Nigeria
3 AI Africa Lab, Lagos 101017, Nigeria
4 School of Public Policy and Global Affairs, University of British Columbia, Vancouver Campus, Vancouver, BC V6T 1Z2, Canada
5 Department of Geography and Resource Management, Chinese University of Hong Kong, Hong Kong, China
* Correspondence: 18042561r@connect.polyu.hk; Tel.: +852-6761-2834

**Abstract:** University towns face many challenges in the 21st century due to urbanization, increased student population, and higher educational institutions' inability to house all their students on-campus. For university towns to be resilient and sustainable, the challenges facing them must be assessed and addressed. To carry out community resilience assessments, this study adopted a novel methodological framework to harness the power of artificial intelligence and social media big data (user-generated content on Twitter) to carry out remote studies in six university towns on six continents using Text Mining, Machine Learning, and Natural Language Processing. Cultural, social, physical, economic, and institutional and governance community challenges were identified and analyzed from the historical big data and validated using an online expert survey. This study gives a global overview of the challenges university towns experience due to studentification and shows that artificial intelligence can provide an easy, cheap, and more accurate way of conducting community resilience assessments in urban communities. The study also contributes to knowledge of research in the new normal by proving that longitudinal studies can be completed remotely.

**Keywords:** machine learning; natural language processing; text mining; social media; studentification; sustainability

## 1. Introduction

As the world experiences geometric growth in population and youth bulge in the 21st century, radical changes had to be made to higher education funding in most countries to meet the increasing demand for university education [1,2]. In most countries, such as the United Kingdom and the United States, these changes have also led to a shift in the funding of most Higher Educational Institutions (HEIs) away from the state, which increased the marketization of higher education [1,3]. According to Brooks, Byford, and Sela [1], the United Kingdom's commercialization of higher education has changed the narratives. Students now "see degrees as private investments rather than public good". To obtain the best "investment", students now travel far away from home in search of "quality" when making their higher education choices. Related to this, Kinton, Smith, Harrison, and Culora [2] emphasised that global competition among HEIs for student "customers" have made universities more responsive, increased their teaching quality and focused on providing more conducive learning environments. For students, framing "students-as-consumers" clearly extends beyond the selection of universities and courses to other aspects of university life, such as residential decision-making, cost of living and

students' lifestyle. As a result of the above, there has been a growing global debate on the changing trends of student geographies. Housing developments are changing from traditional living pathways (on-campus accommodation) to off-campus shared Housing with Multiple Occupancies (HMOs) and Purpose-Built Students Accommodation (PBSA) enclaves, which gradually change the morphology of university towns and affect their sustainability [2,4,5].

"Studentification", a term coined by British geographer Darren P. Smith in 2002, has been globally used to describe the significant processes of urban change and the challenges university towns face due to the growing students' concentration off-campus. This is due to the inability of universities to house all their students within their campuses [4,6–8]. Some of the impacts of studentification have been well documented in the research corpus for the last two decades, but they were mainly woven around housing studies. Hence, most existing studies mainly discuss the economic, social, and environmental negative impacts of housing and students' accommodation and proffer solutions around the same issues using human geography and social theories [2,9–15]. For university towns to be sustainable, they have to be resilient against the chronic stresses and shocks affecting them [16]. Building resilience requires a holistic assessment in all the dimensions of resilience [17,18]. Review of extant studentification literature shows that there are no studies looking at the negative impacts of studentification from the community resilience perspective, providing holistic community assessment, or identifying community challenges from textual big data using artificial intelligence [19].

To fill this identified research gap, this study proposed a novel Community Resilience Assessment (CRA) framework that uses Artificial Intelligence (AI) tools to identify and holistically assess community challenges within university towns. The research answered the questions of the possibility of using AI and textual big data to assess community challenges and the reliability of using such an assessment in university towns suffering from the negative impacts of studentification. We chose six university towns as case studies. Namely: Loughborough in Leicestershire, UK; Akoka in Lagos, Nigeria; Ann Arbor in Michigan, USA; Hung Hom in Kowloon, Hong Kong; Sydney in New South Wales, Australia; and Aguita de la Perdiz in Concepcion, Chile. These towns were selected because they have the highest studentification user-generated content in each continent based on Twitter's big data. Figure 1 shows the geo-location of the six case studies.

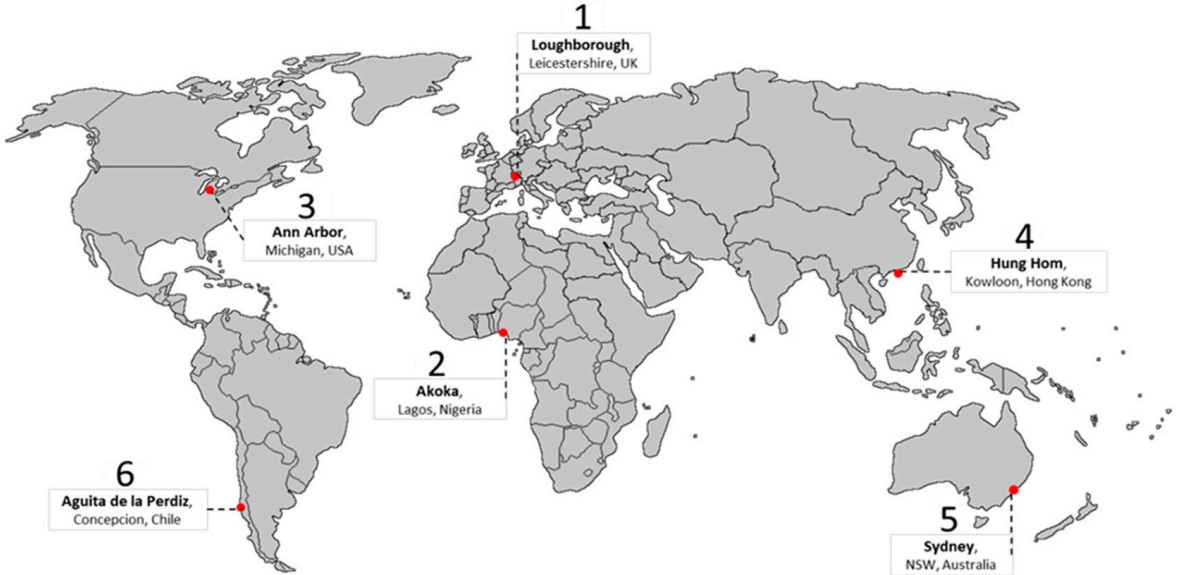

**Figure 1.** Map showing the location of the six case studies. Source: Authors' fieldwork.

This study gives a global overview of university towns' challenges due to studentification beyond the housing issues often discussed in the literature. It also shows that AI and textual big data from microblogs can provide an easy, cheap, and more accurate way of conducting community resilience assessments. Section 2 of this paper shows the literature review and other related work, Section 3 explains the methodology, Section 4 shows the results from the case studies, Section 5 discusses the findings, and Section 6 gives the summary and conclusion as well as the limitations and areas for future research.

## 2. Theoretical and Conceptual Background

### 2.1. Studentification: Practical Challenges and Benefits

Studentification leads to urban changes over time. According to Smith [20] and Situmorang et al. [21] these changes have five key dimensions: social, cultural, physical, economic, and governance. Socially, studentification leads to structural gentrification and segregation. Culturally, the social clusters or concentrations of youths with shared students' culture, lifestyle, and consumption practices lead to the introduction of new sub-cultures in the area. Physically, the environment may either be upgraded to cater to the new teaming customers (especially in retail and service infrastructure) or downgraded to a slum over time. And economically, housing stock changes to accommodate the student population lead to higher densities and inflation of property and rental prices. Local businesses also change their models over time to satisfy the needs of the students. With such rapid new complexities in the university towns, governance issues gradually manifest.

Although studentification is often portrayed as a negative phenomenon in the media and research, the town-gown relationship is not all parasitic. Some of the benefits of studentification to the university towns and their residents include the following: the provision of a young and educated workforce, cheaper labour and increased volunteerism [22]; adding more diversity and vibrancy to local cultures and raising the aspirations of the local youths [23]; enhancing the spending power, improving the local economy, creating more jobs and sustaining the local retail businesses [24]; supporting the local real estate sector and its associated trades (agency, insurance, finance, etc.) and driving up demands for quality housing provision [25]; as well as making the town more attractive to tourists and investors [26]. However, this study only looks at the practical challenges studentification has on university towns and their residents.

### 2.2. The Concept of Sustainability, Resilience, and Community Resilience Assessment

Defining sustainability depends on the framing and dimension. A common framework with substantial nexus with resilience is "the triple bottom line", which conceptualizes that societies should not make decisions about their future based only on economic returns but also on environmental protection, social justice, and equity [27]. The principle of the triple bottom line suggests that human settlements must be environmentally bearable, socially equitable, and economically viable for the current generations and the future ones yet unborn [28]. According to UN-Habitat [29], resilience is essential to sustainability. That is why United Nations Sustainable Development Goal 11 (UNSDG 11) categorically mandated the 193 UN member nations to strive to make their human settlements inclusive, safe, resilient, and sustainable. In urban planning, the "*concept of resilience*" is defined as the ability of human settlements to prepare and plan for, absorb, recover from, and more successfully adapt to environmental, social, and economic adverse events [30]. Community resilience, therefore, is learning from the past, understanding current situations and using that information to minimize future negative impacts. Influenced by the above philosophy and the global call to develop a sustainable world, as well as the increasing challenges of human settlements, resilience research and the concept of community resilience assessment are fast becoming popular in global policy and scientific research and discourse [31].

Community Resilience Assessment (CRA) is an assessment carried out to identify and analyze the challenges human communities face [32]. CRAs are summative or formative toolkits, indexes, scorecards, and frameworks that identify and analyze socio-cultural, eco-

nomic, environmental, and institutional community resilience challenges [31]. Sharifi [31] posited that good CRA methodologies should be able to identify community challenges in all dimensions of resilience, capture spatiotemporal dynamism, address uncertainties, and seek the opinions of the people involved. In the last two decades, more than 100 CRA methodologies (toolkits, indexes, scorecards, and frameworks) have been created by different organizations for different purposes, countries, or regions. No CRA methodology was explicitly developed to identify or assess community challenges in university towns. However, few can be modified to identify and evaluate specific challenges within university towns, such as natural disasters and climate change impacts.

*2.3. The Use of Artificial Intelligence and User-Generated Content from Social Media Microblogs in Community Resilience Assessment*

Processes in the built environment have seen a lot of disruptions in the 21st century [33]. This is mainly due to the new challenges human settlements face in the 21st century, coupled with the drive for smarter cities, the widespread use of AI, and the explosive data generation in the fourth industrial revolution [34]. Today, billions of data points are generated in cities globally because of the increase in internet usage and smart gadgets (Internet of Things) [35]. The rising complexities and challenges of our cities in this information age require new innovative methods because most traditional approaches can no longer harness the potential of the big data generated in our cities [36]. To rise to the occasion, professionals and researchers in the built environment now use AI systems to automate traditional processes and make them more efficient and smarter [37].

In simple terms, the vast and constantly expanding field of AI refers to machines or computers mimicking cognitive functions that humans associate with the human mind, such as learning and solving problems [38]. AI applications are being used in almost every sector. In urban planning, AI is used in security surveillance and smart transport systems (including traffic management) [39], robotics, automation and installation of infrastructure [40], health care delivery [40], garbage collection [41], air quality monitoring [42], and disaster management [43], among others. On the other hand, Machine Learning (ML) is a subfield of AI that trains machines to learn from experiences and make intelligent decisions with or without supervision [44]. One of such functions is learning human languages, communicating with humans, and reading human emotions [45]. This subfield of ML is called Natural Language Processing (NLP). Figure 2 summarizes the AI, ML, and NLP relationships.

Social media microblogs have become a key medium of communication and expression with the increased use of Internet of Things (IoT) and smartphones. This has made User-Generated Content (UGC) from Twitter, WeChat, Facebook, and Instagram a huge part of research in areas such as marketing, commerce, tourism, and health [46]. For example, Alharbi et al. [47] used Twitter big data, ML, and NLP methods to study the opinions of Apple phone users. Their research examines users' sentiments to determine if they are happy or sad about using the new iPhones. Using a similar methodology and Twitter big data, Asghar et al. [48] also studied people's automobile preferences. Generally, in commerce and marketing, companies use UGC to understand customers' perceptions and satisfaction and how their goods and services are compared with other similar products in the market [49].

In the health and human settlements nexus, Carlos et al. [50] used Twitter data to study the outbreaks of dengue fever in Brazil, while Shah et al. [51] used data from medical microblogs to analyse the sentiments patients have toward their physicians in the UK. And in travel and tourism, Nilashi et al. [52] used data from social media microblogs and ML to study travellers' decision-making processes and develop a system to recommend hotels tailored to their preferences. Similarly, Sun et al. [53] also used big data from social media to study trends and tourists' opinions in China. Ahani et al. [54] also used a similar methodology to study customer behaviour and customer satisfaction in the hotel industry

to develop a better marketing plan and recommend strategies for hotel owners to increase customer satisfaction and retention.

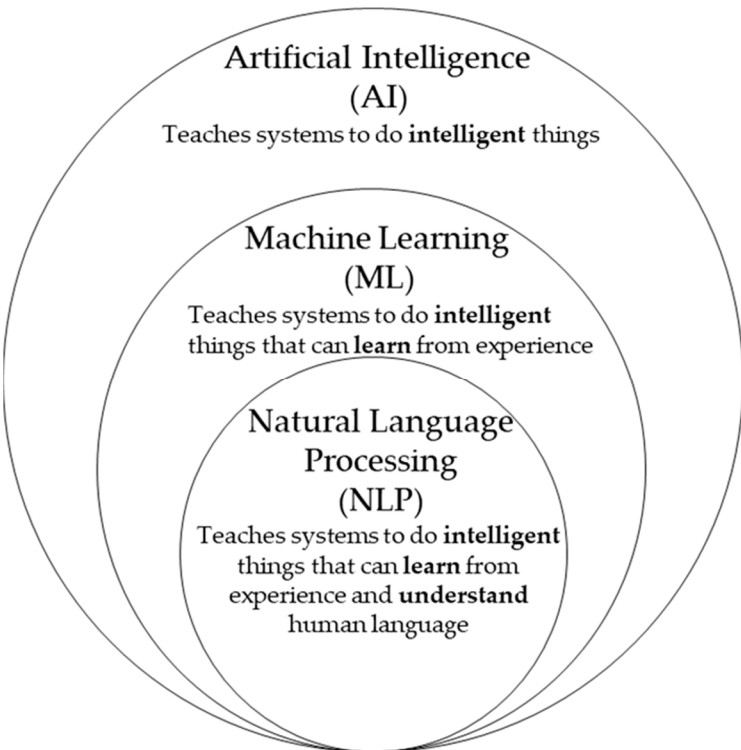

**Figure 2.** The AI—ML—NLP nexus.

In an attempt to use similar methodologies above for urban planning and management Abdul-Rahman et al. [55] developed a framework to simplify pre-processing of social media big data using Text Mining ™, ML, and NLP. Their study showed that UGC from Twitter can be used to identify community challenges using AI. Similar to studies in marketing and tourism, people also share their opinions and sentiments on how they feel about their communities, what challenges their communities experience, and what they think the solutions are. This study expanded Abdul-Rahman, Chan, Wong, Irekponor, and Abdul-Rahman [55] study and methodology to develop a CRA framework for university towns. Apart from its efficiency, the novelty of this proposed framework lies in its ability to provide spatiotemporal analysis of community challenges among the five dimensions of resilience.

Among all social media microblogs, Twitter is commonly used for text mining because of the rich textual UGC, the size of the data, and the ease of using the Twitter API [56]. This study also used big data from Twitter.

### 3. Materials and Methods

Since this study adopted an existing AI-based framework with high accuracy [55], only key modified codes and procedures were repeated here. However, apart from adapting the framework to identify and assess the negative impacts of studentification in multiple case studies, the original approach's validation step was also modified to online experts' validation. This makes validation easier, faster, and cheaper.

The methodological framework in Figure 3 comprises the following steps:

(a)     *Getting started*—The user connects the computer (*Local Host*) to the internet.
(b)     *Connecting to case study and Python environment*—User receives geographical co-ordinates from case study and launches Python v3 (or a newer version) (Python Software Foundation, Beaverton, OR, USA), launches PyQuery, and Lxml.

(c)      *Text mining*—The User downloads the *Optimized-Modified-GetOldTweets3-OMGOT* (https://github.com/marquisvictor/Optimized-Modified-GetOld-Tweets3-OMGOT, accessed on 24 December 2022) library from GitHub and follows the instructions in the *ReadMe file* to mine public UGC from Twitter. *Optimized-Modified-GetOldTweets3-OMGOT* is a python-based open-source tool containing a set of programmatic algorithms designed by Abdul-Rahman, Chan, Wong, Irekponor, and Abdul-Rahman [55] to streamline searches and bypass the rate limits of the Twitter APIs, allowing the download of unlimited historic tweets generated from a specific geo-location using the PyQuery tool, from terminal or command prompt. The algorithms download both the UGC (tweets) and their metadata into Microsoft Excel files (.csv) directly to the *Local Host*. Since the data is downloaded to .csv file(s), it can easily be transferred outside of the Python environment for further data analysis. In this study, only tweets in the English language were downloaded.

(d)      *Topic Modelling*—*Latent Dirichlet Allocation (LDA)* (https://github.com/lda-project/lda, accessed on 24 December 2022) An ML and NLP Python-based tool were used to split the big data downloaded in step (c) into major topics. These topics represent major discussion themes within the selected case study areas based on Twitter UGC. 45 themes (topics) were identified. The 45 topics were converted to keywords and used to re-mine the textual data "per topic" using the *Use Cases* in the *ReadMe file*. Data from each topic was then saved in a separate .csv file. This step helps to validate the previously mined data and break down the big data into manageable sizes for further analysis. Blei et al. [57], Chuang et al. [58], Sievert–Shirley [59], Moody et al. [60], Momtazi [61], Abdul-Rahman, Chan, Wong, Irekponor, and Abdul-Rahman [55] and Asghari, et al. [62] all published great papers on how to use LDA.

(e)      *Sentiments Analysis*—Each topic folder in step (d) was analyzed for sentiment polarity using *Valence Aware Dictionary and sEntiment Reasoner* (VADER) (https://github.com/cjhutto/vaderSentiment, accessed on 24 December 2022). VADER is an ML and NLP open-source tool that analyses textual data according to their sentiment polarity (positive, negative, and neutral) and intensity [63]. Negative comments from the community residents and visitors represent displeasure and community challenges. Due to the unstructured nature of the social media data, VADER is one of the best NLP tools for analysing sentiments from social media UGC [47,48].

(f, g and h)      *Survey and Data Validation*—VADER is trained and validated by the developers [64], and Abdul-Rahman, Chan, Wong, Irekponor, and Abdul-Rahman [55] showed that the output has high accuracy. However, to further reduce bias and narrow the error margin, the assumption that the residents, workers, and visitors' displeasures about a community (negative polarities) represent the community's challenges needs to be re-validated. Physical distribution of the questionnaire survey as used by Abdul-Rahman, Chan, Wong, Irekponor, and Abdul-Rahman [55] slows down the process, therefore, this study proposed an online survey via email and twitter to experts identified through research databases and some identified from the big data based on their work on studentification and community resilience, sustainability and artificial intelligence in the 6 countries of the case studies. The survey instrument was designed and tested followed techniques used by Darko [65]. A pilot survey was carried out before the main questionnaire survey. The purpose of the pilot survey was to test the survey procedures and verify the comprehensiveness and the use of technical language [66]. The pilot survey was administered to five participants: two professors, one chief resilience officer, one post-doctoral researcher, and a doctoral researcher. These participants are all well knowledgeable in the field of CRA and the use of artificial intelligence for big data

mining and natural language processing. After the pre-testing phase, the survey instrument was perfected and administered to experts for seven months, from June 2020 to February 2021. The experts were asked to forward the questionnaire link to others they feel are eligible to answer the questionnaire within their network, including experts outside of their countries and copy the research team. A total of 392 valid responses were received. Figure 4 shows the number of responses received for validation and the extra 17 countries the survey snowballed to. The questionnaire used for this study is available online via https://theses.lib.polyu.edu.hk/handle/200/11732 (pg. 99–203), accessed on 24 December 2022. Only sections A and B were used for validation in this study. Section A was used to collect the respondents' biodata. In contrast, section B collected data on the respondents' countries and the respondents' agreements on the data grouped under the five dimensions of studentification (cultural, social, physical, economic, and institutional and governance challenges). A 5-point Likert scale (1 = strongly disagree; 2 = somewhat; disagree; 3 = neither agree nor disagree; 4 = somewhat agree; 5 = strongly agree). Four data analysis methods were used: (1) The reliability of the scales was measured using Cronbach's alpha; (2) Ranking was performed using Mean value ranking; (3) Standard Deviation scores; (4) The Mean values were normalized (Normalized value = (mean—minimum mean)/(maximum mean—minimum mean)). SPSS v26 and Python v3.10.8 were used for the validation analysis.

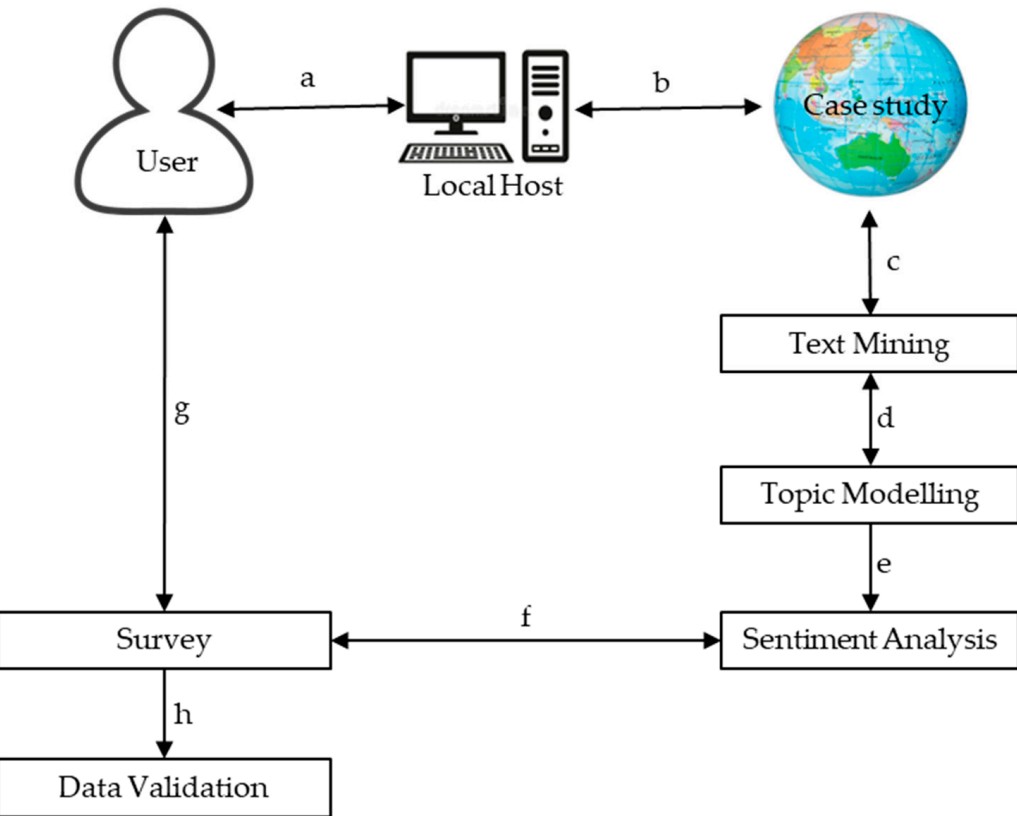

**Figure 3.** The methodological framework adapted from Abdul-Rahman, Chan, Wong, Irekponor, and Abdul-Rahman [55].

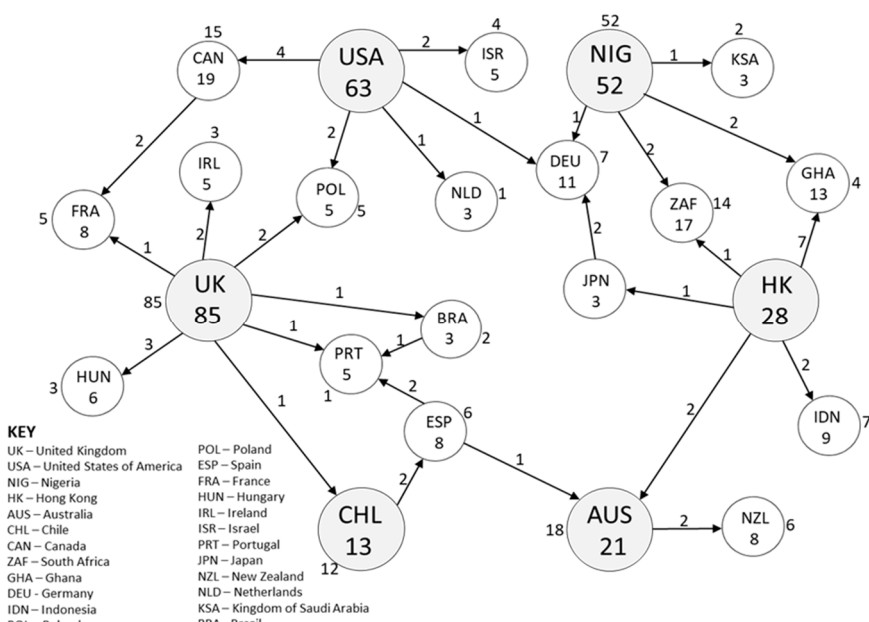

**Figure 4.** Network showing how the questionnaire survey snowballed from the six countries into 23.

## 4. Results

### 4.1. Data Mining using the Optimized-Modified-GetOldTweets3-OMGOT Library

Ten years of Twitter's historic UGC within the six case study areas was downloaded (from 1 January 2010, to 31 December 2020). A total of 4,577,107 tweets containing slags and emojis and their metadata (usernames, permalinks, replies, favourites, dates, etc.) were mined from all case studies. See Table 1 in Supplementary Data for the breakdown of the tweets per case study and Appendix A for the codes used for text mining and data cleaning.

**Table 1.** Case studies and the number of tweets downloaded.

| S/N | Case Study | Number of Tweets (UGC) Mined | |
| | | First Mining | Based on Topics |
|---|---|---|---|
| 1 | Loughborough, UK | 1,297,112 | 1,292,011 |
| 2 | Ann Arbor, USA | 1,052,425 | 1,049,385 |
| 3 | Akoka, Nigeria | 936,575 | 935,822 |
| 4 | Hung Hom, Hong Kong | 724,055 | 721,776 |
| 5 | Sydney, Australia | 502,615 | 498,473 |
| 6 | Aguita de la Perdiz, Chile | 64,325 | 63,844 |
| | **Total** | 4,577,107 | 4,561,311 |

### 4.2. Topic Modelling and Identifying Community Challenges Using Latent Dirichlet Allocation

A total of 45 topics were identified from the first mining datasets combined (total) using LDA. The topic modelling was also performed per case study. A total of 31 of the 45 issues match those from Loughborough's data, 28 from Ann Arbor, 35 from Akoka, 18 from Hung Hom, 22 from Sydney, and 17 from Aguita de la Perdiz. The data mining was then repeated in the case studies based on each topic found in the case studies using case 3 of the *Optimized-Modified-GetOldTweets3-OMGOT* library (see Abdul-Rahman, Chan, Wong, Irekponor and Abdul-Rahman [55]). A total of 4,561,311 tweets were mined under the 45 topics (99.65% of the first mining). A total of 15,796 tweets were automatically excluded because they did not fit into any of the primary 45 topic clusters, and the topics they were under didn't have significant data under them. See Table 2 for the final output and Appendix B for the coding scripts.

**Table 2.** 45 topics generated from the big data mined from the 6 case study areas.

| Theme | Code | Generated Topics | Number of Mined Tweets per Case Study | | | | | |
|---|---|---|---|---|---|---|---|---|
| | | | Lough-Borough | Ann Arbor | Akoka | Hung Hom | Sydney | Aguita de la Perdiz |
| Cultural | C01 | Demographic changes leading to more youths | 30,178 | 21,553 | 8173 | 29,324 | 21,003 | - |
| | C02 | Declining moral and community values | 18,526 | - | - | - | - | 2520 |
| | C03 | Lack of community cohesion & integration due to the transient nature of the student population | 16,124 | - | 8062 | - | 8352 | - |
| | C04 | Aversion of crime and barriers to community policing caused by a transient population | - | 22,652 | 18,251 | - | - | - |
| | C05 | Differing standards of acceptable behaviours by different social groups | - | - | 12,335 | - | - | 2992 |
| | C06 | Cultural diversity and lifestyle conflicts | 15,251 | 25,872 | - | 48,764 | - | - |
| | C07 | Divergent perceptions on what makes up communal obligations | - | - | 10,072 | - | 7983 | - |
| | C08 | Inconsideration and lack of place attachment | 21,261 | 17,008 | 8586 | - | - | - |
| | C09 | Increased racism, tribalism and religious challenges | - | - | 10,611 | - | - | - |
| Social | S01 | Increased anti-social behaviour and social disorder. | 116,352 | 72,555 | 70,055 | - | 40,021 | - |
| | S02 | High level of crime due to the vulnerability & carelessness of the youthful population | - | 26,881 | 9356 | 27,013 | - | 4144 |
| | S03 | Increased level of alcoholism, drugs peddling and abuse. | 65,444 | 57,637 | 47,014 | 17,271 | 25,551 | 4252 |
| | S04 | Increased level of prostitution and sexually transmitted diseases | - | - | 42,625 | - | - | - |
| | S05 | Loss of social services such as reduction in catchment areas for public schools & elderly care | 15,009 | 27,321 | - | - | - | - |
| | S06 | Marginalization of permanent residents | - | 30,764 | - | - | 18,562 | - |
| | S07 | Displacement/replacement of established residents (gentrification) | 18,111 | 50,002 | 18,152 | 26,962 | 39,623 | 4592 |
| | S08 | Increased competition for privately rented apartments | 14,889 | 31,666 | 9176 | - | 7063 | - |
| | S09 | Lack of year-round goods & services due to the resort-economy nature of the community | - | 14,414 | 8003 | - | - | - |

**Table 2.** *Cont.*

| Theme | Code | Generated Topics | Number of Mined Tweets per Case Study | | | | | |
| | | | Lough-Borough | Ann Arbor | Akoka | Hung Hom | Sydney | Aguita de la Perdiz |
|---|---|---|---|---|---|---|---|---|
| | S10 | Establishments of night-time entertainment ventures at the detrimental impacts of residential amenities | 14,752 | 33,111 | 17,787 | - | 6994 | - |
| | S11 | Segregation and social stratification | - | 17,526 | 11,773 | 39,563 | - | - |
| | S12 | Lack of social interactions among groups | - | - | - | 51,033 | - | - |
| Physical | P01 | Illegal subdivision of family homes & apartments into housing with multiple occupancies | 142,858 | 100,369 | 88,426 | 51,723 | 50,522 | 6771 |
| | P02 | Changes in community land use | 21,016 | 16,336 | 34,795 | - | - | - |
| | P03 | Community slumification due to the decline in housing renovations and environmental maintenance. | 71,003 | 42,732 | 25,892 | 16,046 | 5627 | 5251 |
| | P04 | Defacing neighbourhoods with graffiti, posters, writings and rental boards and advertisements | 91,251 | 86,375 | 16,251 | 41,324 | 29,351 | 5931 |
| | P05 | Congestion and overcrowding on the streets and in public places including shops. | - | - | 13,998 | 34,883 | 12,413 | 2221 |
| | P06 | Increased population density | 66,521 | 46,788 | 9005 | - | 32,102 | - |
| | P07 | High environmental pollution—Noise, air pollution and indiscriminate waste/garbage disposal | 100,526 | 74,576 | 58,524 | 89,261 | 52,061 | 7220 |
| | P08 | Increased incidents of protests leading to vandalism of the physical environment. | - | - | 9222 | 91,222 | - | - |
| | P09 | Increased pressure on urban basic services due to higher population than planned for | 17,653 | 10,169 | - | - | 5165 | - |
| | P10 | On-street parking and traffic congestion | 74,251 | 34,001 | - | 25,421 | 14,006 | - |
| | P11 | Pressure on public transport | - | - | 51,196 | - | - | 1942 |
| | P12 | Ghost community during off-term periods | 11,993 | - | 20,014 | - | - | 2014 |

**Table 2.** *Cont.*

| Theme | Code | Generated Topics | Number of Mined Tweets per Case Study | | | | | |
|---|---|---|---|---|---|---|---|---|
| | | | Lough-Borough | Ann Arbor | Akoka | Hung Hom | Sydney | Aguita de la Perdiz |
| Economic | E01 | High rental prices | 95,267 | 99,761 | 81,153 | 45,999 | 47,002 | 5032 |
| | E02 | Lucrative student housing business deters access to affordable housing for non-student residents. | 11,782 | - | 15,551 | - | - | - |
| | E03 | Change in consumer behaviour & taste leading to changes in business models & structures. | 44,031 | 16,094 | 23,623 | 23,061 | 5026 | - |
| | E04 | High cost of living (goods and services) | 57,220 | 39,691 | 76,011 | 35,752 | 43,873 | 4803 |
| | E05 | High influx of commercial activities | 40,308 | 11,452 | 29,112 | 27,154 | 16,021 | 1701 |
| | E06 | Seasonal demand for students' accommodation | 11,506 | - | - | - | 10,152 | - |
| | E07 | Seasonal scarcity of manpower in shops, restaurants, bars, etc. | 13,991 | - | - | - | - | 1441 |
| | E08 | Seasonal customer base (on and off term periods) | 12,016 | - | 9937 | - | - | - |
| | E09 | Low tax generation from the community since students are exempted from taxation. | 35,478 | - | - | - | - | 1017 |
| Institution & Governance | I01 | Weak and disjointed community leadership | - | 12,007 | 38,927 | - | - | - |
| | I02 | Neglect by politicians due to low voting power. | 14,666 | - | 15,261 | - | - | - |
| | I03 | Challenges to existing urban plans and policies | 12,777 | 10,072 | 8893 | - | - | - |
| **Total Tweets** | | | 1,292,011 | 1,049,385 | 935,822 | 721,776 | 498,473 | 63,844 |
| **No of Topics** | | | 31 | 28 | 35 | 18 | 22 | 17 |

*4.3. Sentiments Analysis Using VADER*

Each tweet within each topic was analysed and classified using the sentiment in­dex in Table 3. Generally, tweets with sentiment matric scores of 0.674 (67%) are re­garded as positive. This means the authors (residents or visitors) are satisfied with the situation in the community. Tweets with scores of 0.0326 (33%) are recorded as neutral, meaning the authors (residents and visitors) are indifferent about the situation. On the other hand, tweets with 0.000 scores are negative and represent complaints or displeasure from residents and visitors [63]. The three scores sum up to 1. For better accuracy, the standardized compound matric scores (sums of all the lexicon ratings) are normalized between −1 and +1 [64]. This means = or >0.05 is a positive sentiment polarity, >−0.05 and <0.05 is neutral, and = or <−0.05 is negative.

**Table 3.** Identified community challenges and their ranks based on the frequency of their negative sentiment polarity from VADER.

| Code | Community Challenges | Frequency (Negative Sentiment Polarity) | Ranking within Case Studies | | | | | | VADER Overall Rank |
| --- | --- | --- | --- | --- | --- | --- | --- | --- | --- |
| | | | Lough-Borough | Ann Arbor | Akoka | Hung Hom | Sydney | Aguita de la Perdiz | |
| P01 | Illegal subdivision of family homes & apartments into housing with multiple occupancies | 381,745 | 1 | 1 | 1 | 3 | 2 | 2 | 1 |
| E01 | High rental prices | 345,156 | 4 | 2 | 2 | 6 | 3 | 5 | 2 |
| P07 | High environmental pollution—Noise, air pollution & indiscriminate waste disposal | 332,071 | 3 | 4 | 5 | 2 | 1 | 1 | 3 |
| S01 | Increased anti-social behaviour and social disorder. | 275,236 | 2 | 5 | 4 | - | 5 | - | 4 |
| E04 | High cost of living (goods and services) | 238,967 | 10 | 9 | 3 | 9 | 4 | 6 | 5 |
| P04 | Defacing neighbourhoods with graffiti, posters, writings & rental boards & advertisements | 223,627 | 5 | 3 | 18 | 7 | 8 | 3 | 6 |
| S03 | Increased level of alcoholism, drugs peddling and abuse. | 189,725 | 9 | 6 | 7 | 17 | 9 | 8 | 7 |
| P03 | Community slumification due to decline in housing renovations & environ. maintenance | 135,996 | 7 | 8 | 12 | 18 | 20 | 4 | 8 |
| S07 | Displacement/replacement of established residents (gentrification) | 125,611 | 18 | 7 | 16 | 14 | 6 | 7 | 9 |
| P10 | On-street parking and traffic congestion | 109,359 | 6 | 11 | - | 15 | 13 | - | 10 |
| P06 | Increased population density | 105,918 | 8 | 10 | 30 | - | 7 | - | 11 |
| E03 | Change in consumer behaviour and taste leading to changes in business models & structures. | 75,403 | 11 | 23 | 13 | 16 | 22 | - | 12 |
| P08 | Increased incidents of protests leading to vandalism of the physical environment. | 61,349 | - | - | 28 | 1 | - | - | 13 |
| E05 | High influx of commercial activities | 57,097 | 12 | 26 | 11 | 12 | 12 | 15 | 14 |
| P02 | Changes in community land use | 56,463 | 16 | 22 | 10 | - | - | - | 15 |
| P11 | Pressure on public transport | 49,726 | - | - | 6 | - | - | 14 | 16 |
| P05 | Congestion and overcrowding on the streets and in public places including shops. | 46,570 | - | - | 21 | 10 | 14 | 12 | 17 |
| S10 | Establishments of night-time ent. ventures at the detrimental impacts of residential amenities | 44,463 | 24 | 12 | 17 | - | 19 | - | 18 |
| C01 | Demographic changes leading to more youths | 44,388 | 14 | 19 | 33 | 11 | 10 | - | 19 |
| S12 | Lack of social interactions among groups | 43,452 | - | - | - | 4 | - | - | 20 |
| I01 | Weak and disjointed community leadership | 37,405 | - | 25 | 9 | - | - | - | 21 |
| S08 | Increased competition for privately rented apartments | 36,104 | 23 | 13 | 29 | - | 18 | - | 22 |
| S06 | Marginalization of permanent residents | 34,667 | - | 14 | - | - | 11 | - | 23 |

**Table 3.** *Cont.*

| Code | Community Challenges | Frequency (Negative Sentiment Polarity) | Ranking within Case Studies | | | | | | VADER Overall Rank |
|---|---|---|---|---|---|---|---|---|---|
| | | | Lough-Borough | Ann Arbor | Akoka | Hung Hom | Sydney | Aguita de la Perdiz | |
| S02 | High level of crime due to the vulnerability & carelessness of the youthful population | 34,497 | - | 16 | 27 | 13 | - | 9 | 24 |
| C08 | Inconsideration and lack of place attachment | 33,932 | 15 | 21 | 32 | | - | - | 25 |
| C06 | Cultural diversity and lifestyle conflicts | 33,427 | 21 | 17 | | 5 | - | - | 26 |
| S11 | Segregation and social stratification | 31,018 | | 20 | 23 | 8 | - | - | 27 |
| E09 | Low tax generation from the community since students are exempted from taxation. | 29,984 | 13 | - | - | - | - | 17 | 28 |
| C04 | Aversion of crime and barriers to community policing caused by a transient population | 29,692 | - | 18 | 15 | - | - | - | 29 |
| S04 | Increased level of prostitution and sexually transmitted diseases | 28,777 | - | - | 8 | - | - | - | 30 |
| C03 | Lack of community cohesion & integration due to the transient nature of the population | 28,061 | 20 | - | 34 | - | 16 | - | 31 |
| P12 | Ghost community during off-term periods | 25,471 | 29 | - | 14 | - | - | 13 | 32 |
| I03 | Challenges to existing urban plans and policies | 24,270 | 27 | 28 | 31 | - | - | - | 33 |
| S05 | Loss of social services such as reduction in catchment areas for public schools, elderly care, etc. | 23,745 | 22 | 15 | - | - | - | - | 34 |
| P09 | Increased pressure on urban basic services due to higher population than planned for | 22,321 | 19 | 27 | - | - | 21 | - | 35 |
| E02 | Lucrative student housing business deters access to affordable housing for non-students | 20,063 | 30 | - | 19 | - | - | - | 36 |
| I02 | Neglect by politicians due to low voting power. | 18,607 | 25 | - | 20 | - | - | - | 37 |
| C02 | Declining moral and community values | 18,497 | 17 | - | - | - | - | 11 | 38 |
| E08 | Seasonal customer base (on and off term periods) | 16,725 | 28 | - | 26 | - | - | - | 39 |
| S09 | Lack of year-round goods & services due to the resort-economy nature of the community | 14,112 | - | 24 | 35 | - | - | - | 40 |
| E06 | Seasonal demand for students' accommodation | 12,415 | 31 | - | - | - | 15 | - | 41 |
| E07 | Seasonal scarcity of manpower in shops, restaurants, bars, etc. | 11,539 | 26 | - | - | - | - | 16 | 42 |
| C09 | Increased racism, tribalism and religious challenges | 8520 | - | - | 24 | - | - | – | 43 |
| C05 | Differing standards of acceptable behaviours by different social groups | 7532 | - | - | 22 | - | - | 10 | 44 |
| C07 | Divergent perceptions on what makes up communal obligations | 7392 | - | - | 25 | - | 17 | - | 45 |

Within each of the identified topics in each case study, there were positive, neutral, and negative UGC tweets. Table A1 in Appendix D contains the summations of all normalized and weighted composite scores (sentiment polarity) for each topic. Table 3 shows the identified community challenges and their ranks based on the frequency of their negative sentiment polarity. While Figures 4–6 show the sentiments polarities in each case study, the thematic cluster of community challenges and the intensity of community challenges in each case study, respectively.

The codes used for the VADER sentiment analysis are also contained in Appendix C. See Hutto and Gilbert [63] for more information on the parameters and scoring of the VADER model on Python.

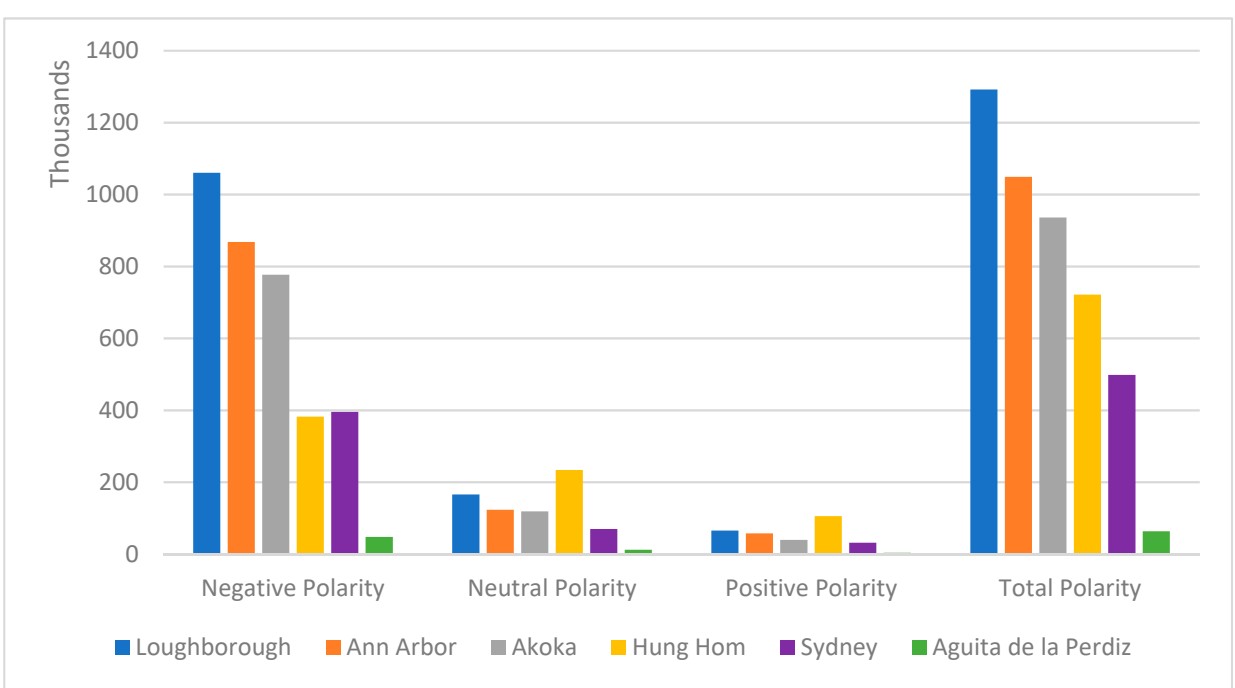

**Figure 5.** Sentiment polarities calculated from the Normalized Weighted Composite Scores (NWCS).

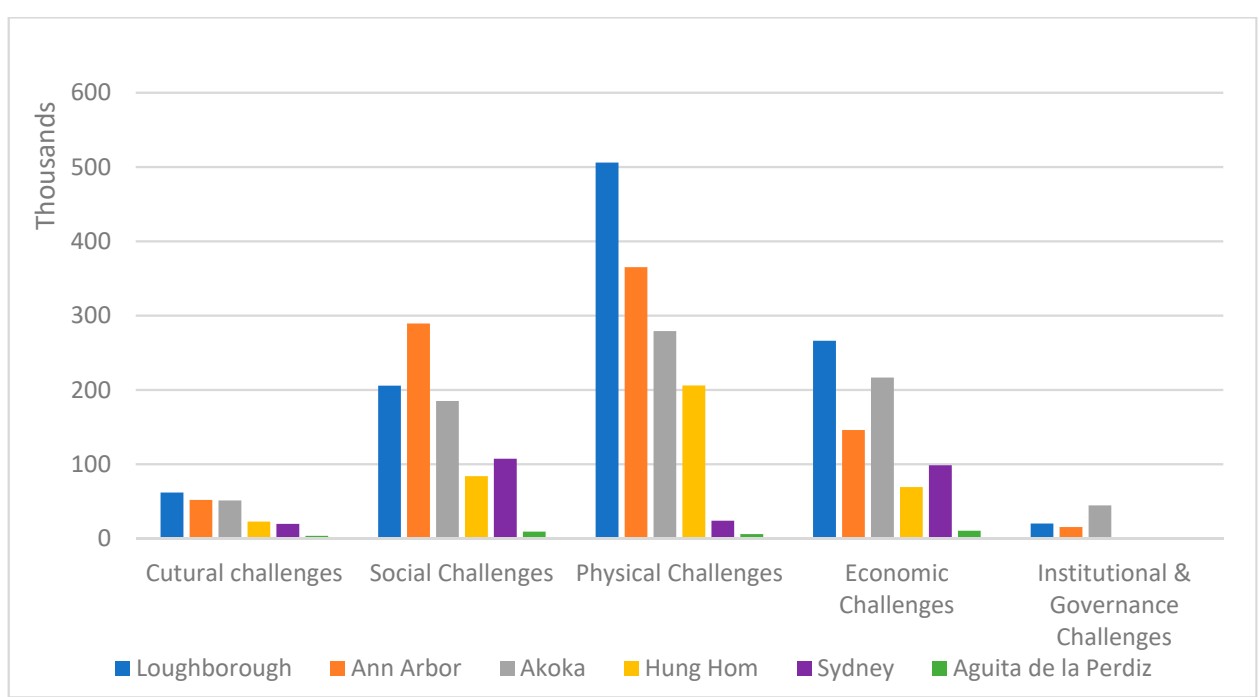

**Figure 6.** Thematic clusters of community challenges in university towns.

*4.4. Result Validation*

To test the reliability of the scales, Cronbach's Alpha (CA) was calculated using Howard [67] Python methodology. The CA values for the subscales were 0.799 (cultural), 0.972 (social), 0.957 (physical), 0.869 (economic), and 0.798 (institutional and governance). By statistical standards, CA scores above 0.7 are said to have good internal consistency [68]; therefore, the validation data is reliable. Table 4 shows the respondents' profile, while the mean values, standard deviation scores, normalized mean values, and ranking of all community challenges are shown in Table 5. All the mean and normalized mean

values were more than the 3.5 and 0.5 average [65], respectively. This means none of the 45 community challenges was collectively rejected by the 392 experts, who were mainly from academia or research institutes and had more than 5 years of experience working as researchers, urban planners, or in the community resilience domain. The majority of the experts also have experience either developing CRA methodology or using one.

**Table 4.** Respondents' profiles for validation.

| Data on Survey Respondents | Responses | Percentage |
|---|---|---|
| **Category** | | |
| Academia/research institute | 189 | 48.2 |
| Consulting/private sector | 42 | 10.7 |
| Public sector/government agency or department | 36 | 9.2 |
| Intergovernmental organization/international NGO | 97 | 24.8 |
| Others | 28 | 7.1 |
| **Profession** | | |
| Academic/researcher | 128 | 32.7 |
| Urban planner | 112 | 28.6 |
| Resilience project manager/officer | 51 | 13.0 |
| Architect | 29 | 7.4 |
| Economist/development economist | 12 | 3.0 |
| Sociologist | 22 | 5.6 |
| Engineer (civil, construction, etc.) | 27 | 6.9 |
| Others | 11 | 2.8 |
| **Years of experience** | | |
| 1–5 years | 36 | 9.2 |
| 6–10 years | 91 | 23.2 |
| 11–15 years | 102 | 26.0 |
| 16–20 years | 55 | 14.0 |
| Above 20 years | 108 | 27.6 |
| **Type of involvement in community resilience & Sustainability** | | |
| Development of as assessment methodology | 191 | 48.7 |
| Use of an assessment method | 138 | 35.2 |
| Both of the above | 51 | 13.0 |
| Others | 12 | 3.1 |

**Table 5.** Validated and ranked community challenges in university towns.

| Code | Community Challenges | VADER Overall Rank | Ranking by Experts in all 23 Countries | | | | Ranking by Experts in the 6 Countries | | | |
|------|----------------------|-------------------|------------|--------------------|------------------------|------|------------|--------------------|------------------------|------|
| | | | Mean Value | Standard Deviation | Normalized Mean Value | Rank | Mean Value | Standard Deviation | Normalized Mean Value | Rank |
| P01 | Illegal subdivision of family homes & apartments into housing with multiple occupancies | 1 | 4.172 | 1.241 | 0.976 | 2 | 4.190 | 1.062 | 0.998 | 3 |
| E01 | High rental prices | 2 | 4.186 | 0.962 | 1.000 | 1 | 4.191 | 0.224 | 1.000 | 1 |
| P07 | High environmental pollution—Noise, air pollution and indiscriminate waste/garbage disposal | 3 | 4.156 | 0.921 | 0.949 | 5 | 4.190 | 0.862 | 0.998 | 2 |
| S01 | Increased anti-social behaviour and social disorder. | 4 | 4.160 | 1.231 | 0.956 | 3 | 4.158 | 0.251 | 0.945 | 6 |
| E04 | High cost of living (goods and services) | 5 | 4.156 | 0.288 | 0.949 | 4 | 4.181 | 0.413 | 0.983 | 4 |
| P04 | Defacing neighbourhoods with graffiti, posters, writings and rental boards and advertisements | 6 | 4.149 | 0.081 | 0.937 | 7 | 4.140 | 0.613 | 0.915 | 8 |
| S03 | Increased level of alcoholism, drugs peddling and abuse | 7 | 4.147 | 0.112 | 0.934 | 9 | 4.131 | 0.251 | 0.900 | 10 |
| P03 | Community slumification due to the decline in housing renovations and environmental maintenance | 8 | 4.152 | 0.177 | 0.942 | 6 | 4.173 | 0.571 | 0.970 | 5 |
| S07 | Displacement/replacement of established residents (gentrification) | 9 | 4.141 | 0.167 | 0.924 | 10 | 4.135 | 0.155 | 0.907 | 9 |
| P10 | On-street parking and traffic congestion | 10 | 4.149 | 0.231 | 0.937 | 8 | 4.141 | 0.352 | 0.917 | 7 |
| P06 | Increased population density | 11 | 4.132 | 1.003 | 0.908 | 13 | 4.122 | 1.216 | 0.885 | 12 |
| E03 | Change in consumer behaviour and taste leading to changes in business models & structures. | 12 | 4.119 | 0.315 | 0.886 | 17 | 4.128 | 1.008 | 0.895 | 11 |
| P08 | Increased incidents of protests leading to vandalism of the physical environment. | 13 | 4.101 | 0.432 | 0.856 | 20 | 4.093 | 0.251 | 0.837 | 17 |
| E05 | High influx of commercial activities | 14 | 4.139 | 0.152 | 0.920 | 11 | 4.115 | 0.624 | 0.874 | 13 |
| P02 | Changes in community land use | 15 | 4.129 | 1.085 | 0.903 | 14 | 4.100 | 0.263 | 0.849 | 15 |
| P11 | Pressure on public transport | 16 | 4.125 | 0.155 | 0.896 | 15 | 4.109 | 0.213 | 0.864 | 14 |
| P05 | Congestion and overcrowding on the streets and in public places including shops. | 17 | 4.135 | 0.262 | 0.913 | 12 | 4.096 | 0.362 | 0.842 | 16 |
| S10 | Establishments of night-time entertainment ventures at the detrimental impacts of residential amenities | 18 | 4.112 | 0.332 | 0.874 | 18 | 4.087 | 1.201 | 0.827 | 19 |
| C01 | Demographic changes leading to more youths | 19 | 4.122 | 0.421 | 0.891 | 16 | 4.081 | 0.521 | 0.817 | 20 |
| S12 | Lack of social interactions among groups | 20 | 4.112 | 1.025 | 0.874 | 19 | 4.090 | 0.241 | 0.832 | 18 |
| I01 | Weak and disjointed community leadership | 21 | 4.084 | 1.045 | 0.827 | 25 | 4.055 | 0.914 | 0.774 | 28 |
| S08 | Increased competition for privately rented apartments | 22 | 4.090 | 0.128 | 0.837 | 23 | 4.069 | 0.269 | 0.797 | 24 |
| S06 | Marginalization of permanent residents | 23 | 4.055 | 0.261 | 0.778 | 30 | 4.079 | 0.323 | 0.814 | 21 |
| S02 | High level of crime due to the vulnerability & carelessness of the youthful population | 24 | 4.058 | 0.383 | 0.783 | 29 | 4.058 | 0.824 | 0.779 | 27 |

**Table 5.** *Cont.*

| Code | Community Challenges | VADER Overall Rank | Ranking by Experts in all 23 Countries | | | | Ranking by Experts in the 6 Countries | | | |
|------|---------------------|--------------------|-------------|--------------------|-------------------------|------|-------------|--------------------|-------------------------|------|
| | | | Mean Value | Standard Deviation | Normalized Mean Value | Rank | Mean Value | Standard Deviation | Normalized Mean Value | Rank |
| C08 | Inconsideration and lack of place attachment | 25 | 4.081 | 0.056 | 0.822 | 26 | 4.061 | 0.731 | 0.784 | 26 |
| C06 | Cultural diversity and lifestyle conflicts | 26 | 4.087 | 0.199 | 0.832 | 24 | 4.075 | 0.518 | 0.807 | 22 |
| S11 | Segregation and social stratification | 27 | 4.099 | 1.074 | 0.852 | 21 | 4.070 | 0.419 | 0.799 | 23 |
| E09 | Low tax generation from the community since students are exempted from taxation. | 28 | 4.091 | 0.361 | 0.839 | 22 | 4.046 | 0.982 | 0.759 | 31 |
| C04 | Aversion of crime and barriers to community policing caused by a transient population | 29 | 4.040 | 1.042 | 0.752 | 33 | 4.050 | 1.043 | 0.766 | 30 |
| S04 | Increased level of prostitution and sexually transmitted diseases | 30 | 4.031 | 1.427 | 0.737 | 34 | 4.041 | 1.099 | 0.751 | 32 |
| C03 | Lack of community cohesion and integration due to the transient nature of the student population | 31 | 4.053 | 1.054 | 0.774 | 31 | 4.051 | 1.011 | 0.767 | 29 |
| P12 | Ghost community during off-term periods | 32 | 4.077 | 1.118 | 0.815 | 27 | 4.063 | 1.231 | 0.787 | 25 |
| I03 | Challenges to existing urban plans and policies | 33 | 4.069 | 1.226 | 0.801 | 28 | 4.019 | 1.306 | 0.714 | 40 |
| S05 | Loss of social services such as reduction in catchment areas for public schools, elderly care, etc. | 34 | 4.011 | 1.118 | 0.703 | 36 | 4.038 | 1.082 | 0.746 | 34 |
| P09 | Increased pressure on urban basic services due to higher population than planned for | 35 | 4.043 | 1.301 | 0.757 | 32 | 4.038 | 1.055 | 0.746 | 33 |
| E02 | Lucrative student housing business deters access to affordable housing for non-student residents. | 36 | 4.011 | 1.230 | 0.703 | 38 | 4.027 | 1.070 | 0.728 | 38 |
| I02 | Neglect by politicians due to low voting power. | 37 | 4.027 | 1.377 | 0.730 | 35 | 4.027 | 1.103 | 0.728 | 39 |
| C02 | Declining moral and community values | 38 | 3.983 | 1.401 | 0.655 | 41 | 4.029 | 1.190 | 0.731 | 37 |
| E08 | Seasonal customer base (on and off term periods) | 39 | 4.008 | 1.231 | 0.698 | 39 | 3.899 | 1.222 | 0.515 | 43 |
| S09 | Lack of year-round goods & services due to the resort-economy nature of the community | 40 | 3.952 | 1.001 | 0.603 | 42 | 4.034 | 1.026 | 0.739 | 36 |
| E06 | Seasonal demand for students' accommodation | 41 | 3.952 | 1.007 | 0.603 | 43 | 4.007 | 1.009 | 0.694 | 41 |
| E07 | Seasonal scarcity of manpower in shops, restaurants, bars, etc. | 42 | 4.011 | 1.180 | 0.703 | 37 | 4.038 | 1.231 | 0.746 | 35 |
| C09 | Increased racism, tribalism and religious challenges | 43 | 3.990 | 1.220 | 0.667 | 40 | 3.989 | 1.025 | 0.664 | 42 |
| C05 | Differing standards of acceptable behaviours by different social groups | 44 | 3.597 | 1.153 | 0.000 | 45 | 3.899 | 1.302 | 0.515 | 44 |
| C07 | Divergent perceptions on what makes up communal obligations | 45 | 3.921 | 1.032 | 0.550 | 44 | 3.589 | 1.247 | 0.000 | 45 |

## 5. Discussion

### 5.1. General Overview of Community Challenges in University Towns

The UGC from the six case studies shows that university towns face similar challenges globally. This was confirmed by the experts' validation since none of the community challenges was rejected. Some of the community challenges, such as increased racism, tribalism, and religious challenges (C09) and increased levels of prostitution and sexually transmitted diseases (S04) were unique to only Akoka (Nigeria). At the same time, the lack of social interactions among groups (S12) was unique to only Hung Hom (Hong Kong). The rest of the community challenges were reported in at least two case studies, as seen in Table 2.

Loughborough, with the highest number of mined UGC (see Table 1), has the highest negative polarity (complaints), followed by Ann Arbor, then Akoka, Hung Hom, Sydney, and Aguita de la Perdiz (see Figure 5). But overall, Akoka has the highest number of community challenges (35 challenges), followed by Loughborough (31 challenges), Ann Arbor (28 challenges), Sydney (22 challenges), Hung Hom (18 challenges), and Aguita de la Perdiz (17 challenges). Thematically, the challenges were grouped into cultural, social, physical (environmental), economic, and institutional and governance challenges. Figure 6 shows that most community challenges identified were physical/environmental, followed by social, economic, cultural, and institutional and governance challenges. However, no institutional and governance challenges were identified from the data in Sydney and Aguita de la Perdiz. Figure 7 shows that 47.8% of the community challenges identified in Loughborough were physical/environmental, 25.1% had to do with the community's economy, 19.4% were social, 5.8% were cultural, and only 1.9% of the community challenges were institutional and governance challenges. In Ann Arbor, 42.1% were physical, 33.3% were social, 16.8% were economic, 6% were cultural, and only 1.8% were institutional and governance challenges. Akoka has 35.9% of her identified community challenges as physical, 28% economic, 23.8% social, 6.6% cultural, and 5.7% institutional and governance issues. Hung Hom has more than half of her community challenges (53.9%) as physical, 22% as social, 18.1% as economic, and 6% as institutional and governance-related challenges (due to studentification). Sydney has 43% social challenges, 39.5% economic, 9.6% physical, and 7.9% cultural. Lastly, Aguita de la Perdiz has 36% economic challenges, 31.7% social, 20.3% physical, and 12% cultural.

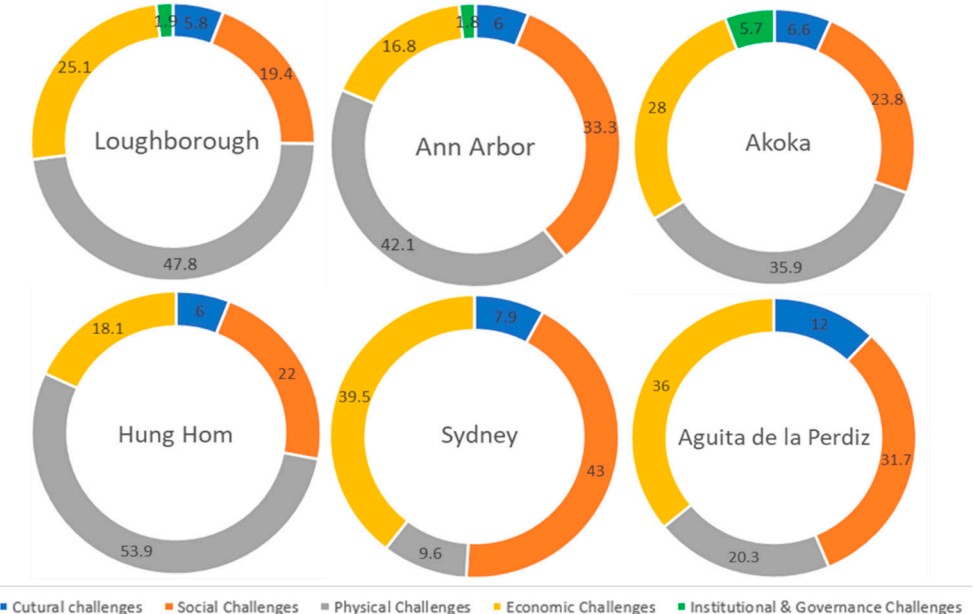

**Figure 7.** Charts showing the intensity of community challenges in percentages in the case studies.

Generally, the overall ranking by the sentiment analyzer (VADER), the ranking by the experts in the 23 countries (total), and those from the 6 case study countries do not differ much. Although the community challenges were ranked slightly differently in the three separate rankings, as shown in Table 5, the top 10 community challenges remain the same across the three rankings. These top 10 community challenges include the following: the illegal subdivision of family homes and apartments into housing with multiple occupancies (P01); high rental prices (E01); high environmental pollution (noise, air pollution and indiscriminate waste/garbage disposal (P07)); increased anti-social behaviour and social disorder (S01); high cost of living (E04); defacing neighbourhoods with graffiti, posters, writings and rental boards and advertisements (P04); increased level of alcoholism, drugs peddling, and abuse (S03); community slumification due to the decline in housing renovations and environmental maintenance (P03); displacement/replacement of established residents (gentrification) (S07); and on-street parking and traffic congestion (P10).

These results show that the intensity of community challenges varies from one community to the other, but overall most university towns experience similar challenges due to studentification. This points to the fact that students have similar behaviours regardless of the country or region [69,70]. This novel CRA framework allows university towns to collaborate and co-produce solutions against studentification challenges, share best practices and learn coping mechanisms from one another, especially those with similar challenges [71].

*5.2. Novelty and Implications of the Proposed CRA Methodology*

a.   Assessment of all major community resilience dimensions

Communities have multiple complex dimensions [72]. This novel framework identified and analysed challenges under the five major dimensions of resilience (cultural, social, physical/environmental, economic, and institution and governance) in all the university towns. This allows community planners and managers to study community resilience challenges holistically and zoom deeper into individual community challenges or resilience dimensions.

b.   Assessing the spatiotemporal dynamism of the community challenges

Capturing time horizons and knowing the specific areas where the residents' and visitors' sentiments were generated will help the community managers better assess the challenges and focus on "hotspots". Since the UGC big data from microblogs such as Twitter come with metadata that contains the date and time of tweets generated within a specified spatial radius, the negative polarities can be modelled further after sentiments analysis using Microsoft Excel 3-D Clustered Columns to show spatiotemporal dynamics. Figure 8 shows a polarity-based model of residents' monthly complaints from 1 January 2010 to 31 December 2020 in Loughborough, UK. The data for P07 (negative sentiments for Loughborough = 98,852 tweets) from figure of Appendix C was grouped into months before it was modelled. The model shows a clear pattern that follows the term periods of Loughborough University and College. The complaints reduced during the summer term and semester three (April to August) and also in December when the university town was almost empty. Over the last 10 years, the complaints about noise and indiscriminate waste disposal have increased in line with the growth of student residents in the town. This model can be generated to analyse any of the community challenges identified.

c.   Addressing uncertainties and ensuring public participation

Carrying out longitudinal studies to understand historical events and analysing patterns help to develop better action plans and reduce uncertainties [73]. This framework gives room for such assessments and provides an opportunity for sampling the opinions of millions of people concerning community issues. The sampled opinions were from residents, workers, and visitors, regardless of gender, race, age, religion, etc.

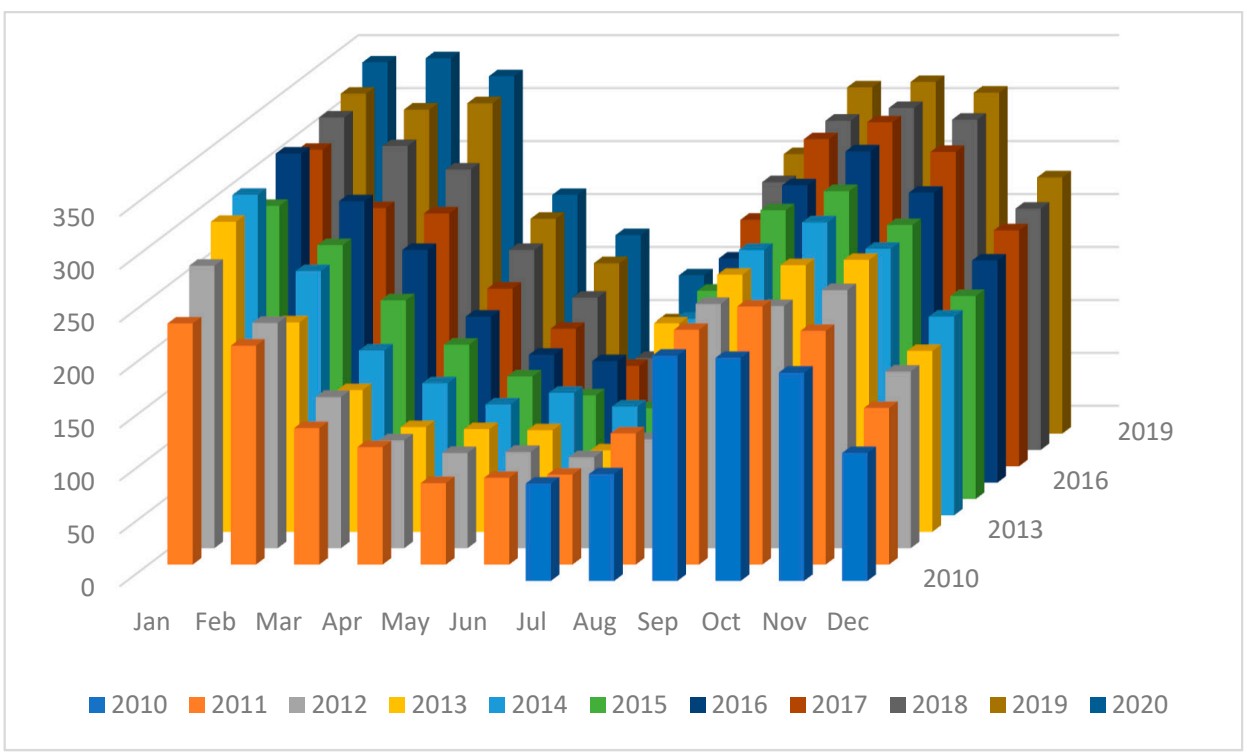

**Figure 8.** Polarity-based model for high environmental pollution (Noise and indiscriminate waste/garbage disposal) in Loughborough, UK.

## 6. Summary and Conclusions

By adapting and modifying the novel framework for pre-processing location-based social media data by Abdul-Rahman, Chan, Wong, Irekponor, and Abdul-Rahman [55], this study demonstrated UGC from microblogs can be used to identify community challenges using AI (ML and NLP) tools such as LDA and VADER. Six university towns were used as case studies.

First, a programmatic algorithm was used to mine the big data using the Twitter API and search engine. Then LDA was used to extract major topics from the data of each case study and the combined big data. These topics were used to re-mine the data, and VADER was used to analyse the sentiment polarity under each issue. The negative Normalized Weighted Composite Scores (NWCS) frequencies were used to rank the identified community challenges. An online expert survey was conducted to validate and rank the negative impacts of studentification. Mean ranking, standard deviation, and normalized mean values were used to rank the community challenges. The statistical results showed that all 45 challenges clustered around the 5 community resilience dimensions were accepted as negative impacts of studentification.

Apart from being comprehensive enough to identify cultural, social, physical/environmental, economic, and institutional and governance challenges in the university towns, this novel framework also provides a deeper spatiotemporal analysis of each community challenge. Using a large opinion poll (sample size) helps minimize errors and increases the accuracy of the data.

This study contributes to the community resilience body of knowledge by providing a simple, fast, cheap, and efficient way of conducting CRA remotely. This novel methodology can be used by urban planners, community managers, community-based organizations, and universities. It can be used to identify community challenges and make university towns resilient and sustainable.

This methodological framework works better in well-connected urban university towns where more people are connected to the internet and the use of social media is high. This limitation will not render the methodology useless, but it will affect the amount of data available for analysis if the framework is used in a rural community with low Internet

connectivity. Future works may include using APIs from other microblogs such as WeChat and Facebook. The framework can also be improved to predict future trends based on historical data. Geographic Information System (GIS) can also be used to overlay the data on the base maps of the case studies to run more analysis and visualization.

**Author Contributions:** Conceptualization, M.A.-R. and Y.A.A.; methodology, M.A.-R., Y.A.A. and O.S.; software, M.A.-R.; validation, W.K.M. and M.I.A.; formal analysis, visualization and data curation, M.A.-R., Y.A.A. and W.K.M.; writing—original draft preparation and writing—review and editing, M.A.-R., O.S. and M.I.A. All authors have read and agreed to the published version of the manuscript.

**Funding:** This research work was part of a larger doctoral study titled "A community Resilience Assessment Framework for University Towns" supported by a PhD studentship from the Research Institute for Sustainable Development (RISUD) and the Department of Building and Real Estate of the Hong Kong Polytechnic University [research grant: G-R006.RJET].

**Institutional Review Board Statement:** Not applicable.

**Informed Consent Statement:** Informed consent was obtained from all respondents involved in the study.

**Data Availability Statement:** Not applicable.

**Acknowledgments:** The authors acknowledge Professor Edwin H.W. Chan and Professor Man Sing Wong's supervision, for their advice and mentorship for the PhD thesis "A community Resilience Assessment Framework for University Towns", which led to the development of this manuscript.

**Conflicts of Interest:** The authors declare no conflict of interest. The funders had no role in the design of the study; in the collection, analyses, or interpretation of data; in the writing of the manuscript; or in the decision to publish the results.

**Appendix A**

1.　　Data mining codes.

```
#First Mining for LDA
python GetOldTweets3.py --near "coordinates from centre of case study" --within
4km --lang es --since 2010-01-01 --until 2020-12-31
```

```
#Second Mining using Keywords for Sentiment Analysis
python GetOldTweets3.py --near " coordinates from centre of case study " --within
4km --lang es --since 2010-01-01 --until 2020-12-31 --querysearch "keywords"
```

2.　　Codes for text cleaning

　　　　a.　　Loading the dataset

```
df = pd.read_csv('file name')
```

　　　　b.　　Data cleaning and noise reduction

```python
from nltk.corpus import stopwords
from nltk.stem.wordnet import WordNetLemmatizer
import string

stop = set(stopwords.words('english'))
exclude = set(string.punctuation)
lemma = WordNetLemmatizer()

def clean(doc):
        """"this is a basic function that takes
    a document as input and cleans it for further use"""

    stop_free = " ".join([i for i in doc.lower().split() if i not in stop])
    punc_free = ''.join(ch for ch in stop_free if ch not in exclude)
    normalized = " ".join(lemma.lemmatize(word) for word in punc_free.split())
    return normalized

doc_clean = [clean(doc).split() for doc in doc_complete]
```

c. Convert the corpus into a document-term matrix.

```python
#Import gensim library
import gensim
from gensim import corpora
#Create the term dictionary for the corpus, where every unique term is assigned
an index.
dictionary = corpora.Dictionary(doc_clean)
#Then, convert the list of documents (corpus) into Document Term Matrix using
dictionary prepared above.
doc_term_matrix = [dictionary.doc2bow(doc) for doc in doc_clean]
```

**Appendix B**

Codes for Topic Modelling using LDA

```python
#Initializing the LDA Model with gensim library
Lda = gensim.models.ldamodel.LdaModel

#Training the LDA model on the document term matrix
ldamodel = Lda(doc_term_matrix, num_topics=50, id2word = dictionary, passes=50)
print(ldamodel.print_topics(num_topics=50, num_words=5))
```

**Appendix C**

Codes for Sentiment Analysis Using VADER

```
#Import VADER library
> pip install vaderSentiment

#Launch the sentiments analyzer
from vaderSentiment.vaderSentiment import SentimentIntensityAnalyzer
analyser = SentimentIntensityAnalyzer()

#Launch the polarity score calculator
def sentiment_analyzer_scores(sentence):
    score = analyser.polarity_scores(sentence)
    print("{:-<40}{}".format(sentence, str(score)))
sentiment_analyzer_scores("Demographic changes leading to more youths.")
The family park is super cool---------------- {'neg': 0.0, 'neu': 0.326, 'pos':
0.674, 'compound': 0.7351}
```

**Appendix D**

**Table A1.** Sentiment polarity for 4,561,311 tweets mined from six university towns in the UK, US, Nigeria, Hong Kong, Australia and Chile from 1 January 2010, to 31 December 2020.

| Case Study | S/N | Topics | negTweets | neuTweets | posTweets | ∑Tweets |
|---|---|---|---|---|---|---|
| | | **Cultural** | | | | |
| | 1 | C01 | 10,524 | 14,756 | 4898 | 30,178 |
| | 2 | C02 | 16,976 | 1492 | 58 | 18,526 |
| | 3 | C03 | 15,026 | 1021 | 77 | 16,124 |
| | 4 | C06 | 5652 | 7635 | 1964 | 15,251 |
| | 5 | C08 | 13,769 | 7184 | 308 | 21,261 |
| | | **Social** | | | | |
| | 6 | S01 | 110,457 | 5715 | 180 | 116,352 |
| | 7 | S03 | 59,820 | 4602 | 1022 | 65,444 |
| | 8 | S05 | 8976 | 4792 | 1241 | 15,009 |
| | 9 | S07 | 10,872 | 6197 | 1042 | 18,111 |
| **Loughborough, UK** | 10 | S08 | 9992 | 2824 | 2073 | 14,889 |
| | 11 | S10 | 5766 | 3624 | 5362 | 14,752 |
| | | **Physical** | | | | |
| | 12 | P01 | 122,825 | 17,924 | 2109 | 142,858 |
| | 13 | P02 | 15,184 | 4625 | 1207 | 21,016 |
| | 14 | P03 | 56,625 | 12,612 | 1766 | 71,003 |
| | 15 | P04 | 78,563 | 11,162 | 1526 | 91,251 |
| | 16 | P06 | 46,021 | 6012 | 14,488 | 66,521 |
| | 17 | P07 | 98,852 | 1645 | 29 | 100,526 |
| | 18 | P09 | 11,524 | 5167 | 962 | 17,653 |
| | 19 | P10 | 68,066 | 5160 | 1025 | 74,251 |
| | 20 | P12 | 8383 | 2186 | 1424 | 11,993 |

**Table A1.** *Cont.*

| Case Study | S/N | Topics | negTweets | neuTweets | posTweets | ∑Tweets |
|---|---|---|---|---|---|---|
| | | **Economic** | | | | |
| | 21 | E01 | 91,251 | 3164 | 852 | 95,267 |
| | 22 | E02 | 9391 | 1526 | 865 | 11,782 |
| | 23 | E03 | 38,726 | 3784 | 1521 | 44,031 |
| | 24 | E04 | 55,692 | 1506 | 22 | 57,220 |
| | 25 | E05 | 13,668 | 11,114 | 15,526 | 40,308 |
| | 26 | E06 | 8644 | 2282 | 580 | 11,506 |
| | 27 | E07 | 10,536 | 3014 | 441 | 13,991 |
| | 28 | E08 | 8904 | 2511 | 601 | 12,016 |
| | 29 | E09 | 29,413 | 5723 | 342 | 35,478 |
| | | **Institution & Governance** | | | | |
| | 30 | I02 | 9725 | 3516 | 1425 | 14,666 |
| | 31 | I03 | 10,526 | 1526 | 725 | 12,777 |
| | **Total** | | **1,060,349** | **166,001** | **65,661** | **1,292,011** |
| | | **Cultural** | | | | |
| | 1 | C01 | 11,241 | 8251 | 2061 | 21,553 |
| | 2 | C04 | 16,340 | 5561 | 751 | 22,652 |
| | 3 | C06 | 11,514 | 12,351 | 2007 | 25,872 |
| | 3 | C08 | 13,005 | 3102 | 901 | 17,008 |
| | | **Social** | | | | |
| | 5 | S01 | 70,045 | 2414 | 96 | 72,555 |
| | 6 | S02 | 20,961 | 4669 | 1251 | 26,881 |
| | 7 | S03 | 53,817 | 2619 | 1201 | 57,637 |
| | 8 | S05 | 14,769 | 9226 | 3326 | 27,321 |
| | 9 | S06 | 23,141 | 5622 | 2001 | 30,764 |
| | 10 | S07 | 48,323 | 1627 | 52 | 50,002 |
| Ann Arbor, USA | 11 | S08 | 16,521 | 9523 | 5622 | 31,666 |
| | 12 | S09 | 9313 | 4098 | 1003 | 14,414 |
| | 13 | S10 | 23,816 | 5783 | 3512 | 33,111 |
| | 14 | S11 | 8784 | 4531 | 4211 | 17,526 |
| | | **Physical** | | | | |
| | 15 | P01 | 97,234 | 2152 | 983 | 100,369 |
| | 16 | P02 | 10,238 | 4242 | 1856 | 16,336 |
| | 17 | P03 | 40,711 | 1400 | 621 | 42,732 |
| | 18 | P04 | 79,518 | 4343 | 2514 | 86,375 |
| | 19 | P06 | 27,825 | 7551 | 11,412 | 46,788 |
| | 20 | P07 | 73,512 | 1008 | 56 | 74,576 |
| | 21 | P09 | 8075 | 1523 | 571 | 10,169 |
| | 22 | P10 | 28,029 | 5161 | 811 | 34,001 |

**Table A1.** *Cont.*

| Case Study | S/N | Topics | negTweets | neuTweets | posTweets | ∑Tweets |
|---|---|---|---|---|---|---|
| | | **Economics** | | | | |
| | 23 | E01 | 92,562 | 5044 | 2155 | 99,761 |
| | 24 | E03 | 11,164 | 3509 | 1421 | 16,094 |
| | 25 | E04 | 36,543 | 2131 | 1017 | 39,691 |
| | 26 | E05 | 5729 | 1217 | 4506 | 11,452 |
| | | **Institution & Governance** | | | | |
| | 27 | I01 | 8674 | 2451 | 882 | 12,007 |
| | 28 | I03 | 6751 | 2328 | 993 | 10,072 |
| | **Total** | | **868,155** | **123,437** | **57,793** | **1,049,385** |
| | | **Cultural** | | | | |
| | 1 | C01 | 4526 | 2643 | 1004 | 8173 |
| | 2 | C03 | 7022 | 1012 | 28 | 8062 |
| | 3 | C04 | 13,352 | 4478 | 421 | 18,251 |
| | 4 | C05 | 5521 | 6104 | 710 | 12,335 |
| | 5 | C07 | 5202 | 3758 | 1112 | 10,072 |
| | 6 | C08 | 7158 | 1395 | 33 | 8586 |
| | 7 | C09 | 8520 | 1241 | 850 | 10,611 |
| | | **Social** | | | | |
| | 8 | S01 | 61,503 | 8109 | 443 | 70,055 |
| | 9 | S02 | 8111 | 733 | 512 | 9356 |
| | 10 | S03 | 44,874 | 2012 | 128 | 47,014 |
| | 11 | S04 | 28,777 | 12,824 | 1024 | 42,625 |
| | 12 | S07 | 11,741 | 5539 | 872 | 18,152 |
| **Akoka, Nigeria** | 13 | S08 | 5545 | 2368 | 1263 | 9176 |
| | 14 | S09 | 4799 | 3000 | 204 | 8003 |
| | 15 | S10 | 11,900 | 3776 | 2111 | 17,787 |
| | 16 | S11 | 8012 | 3652 | 109 | 11,773 |
| | | **Physical** | | | | |
| | 17 | P01 | 79,721 | 2254 | 6451 | 88,426 |
| | 18 | P02 | 31,041 | 1782 | 1972 | 34,795 |
| | 19 | P03 | 18,955 | 6645 | 292 | 25,892 |
| | 20 | P04 | 8563 | 5172 | 2516 | 16,251 |
| | 21 | P05 | 8934 | 4441 | 623 | 13,998 |
| | 22 | P06 | 5662 | 2120 | 1223 | 9005 |
| | 23 | P07 | 57,204 | 1217 | 103 | 58,524 |
| | 24 | P08 | 4726 | 3512 | 984 | 9222 |
| | 25 | P11 | 48,461 | 2583 | 152 | 51,196 |
| | 26 | P12 | 15,965 | 3026 | 1023 | 20,014 |

**Table A1.** *Cont.*

| Case Study | S/N | Topics | negTweets | neuTweets | posTweets | ∑Tweets |
|---|---|---|---|---|---|---|
| | | **Economic** | | | | |
| | 27 | E01 | 79,176 | 1326 | 651 | 81,153 |
| | 28 | E02 | 10,672 | 1231 | 3648 | 15,551 |
| | 29 | E03 | 19,980 | 2641 | 1002 | 23,623 |
| | 30 | E04 | 74,590 | 1320 | 101 | 76,011 |
| | 31 | E05 | 24,432 | 562 | 4118 | 29,112 |
| | 32 | E08 | 7821 | 1085 | 1031 | 9937 |
| | | **Institution & Governance** | | | | |
| | 33 | I01 | 28,731 | 9204 | 992 | 38,927 |
| | 34 | I02 | 8882 | 4516 | 1863 | 15,261 |
| | 35 | I03 | 6993 | 1682 | 218 | 8893 |
| | **Total** | | **777,072** | **118,963** | **39,787** | **935,822** |
| | | **Cultural** | | | | |
| | 1 | C01 | 6632 | 20,571 | 2121 | 29,324 |
| | 2 | C06 | 16,261 | 15,751 | 16,752 | 48,764 |
| | | **Social** | | | | |
| | 3 | S02 | 2301 | 16,304 | 8408 | 27,013 |
| | 4 | S03 | 6015 | 10,502 | 754 | 17,271 |
| | 5 | S07 | 18,027 | 6232 | 2703 | 26,962 |
| | 6 | S11 | 14,222 | 18,150 | 7191 | 39,563 |
| | 7 | S12 | 43,452 | 5798 | 1783 | 51,033 |
| | | **Physical** | | | | |
| **Hung Hom, Hong Kong** | 8 | P01 | 29,522 | 16,025 | 6176 | 51,723 |
| | 9 | P03 | 11,032 | 4002 | 1012 | 16,046 |
| | 10 | P04 | 26,821 | 9991 | 4512 | 41,324 |
| | 11 | P05 | 31,992 | 2016 | 875 | 34,883 |
| | 12 | P07 | 47,885 | 23,653 | 17,723 | 89,261 |
| | 13 | P08 | 56,623 | 18,637 | 15,962 | 91,222 |
| | 14 | P10 | 2162 | 20,015 | 3244 | 25,421 |
| | | **Economic** | | | | |
| | 15 | E01 | 34,112 | 10,681 | 1206 | 45,999 |
| | 16 | E03 | 2572 | 18,618 | 1871 | 23,061 |
| | 17 | E04 | 28,190 | 7074 | 488 | 35,752 |
| | 18 | E05 | 4332 | 9801 | 13,021 | 27,154 |
| | **Total** | | **382,153** | **233,821** | **105,802** | **721,776** |

**Table A1.** *Cont.*

| Case Study | S/N | Topics | negTweets | neuTweets | posTweets | ∑Tweets |
|---|---|---|---|---|---|---|
| | | **Cultural** | | | | |
| | 1 | C01 | 11,465 | 2215 | 7323 | 21,003 |
| | 2 | C03 | 6013 | 2120 | 219 | 8352 |
| | 3 | C07 | 2190 | 4542 | 1251 | 7983 |
| | | **Social** | | | | |
| | 4 | S01 | 33,231 | 6559 | 231 | 40,021 |
| | 5 | S03 | 22,338 | 2451 | 762 | 25,551 |
| | 6 | S06 | 11,526 | 5632 | 1404 | 18,562 |
| | 7 | S07 | 33,243 | 4237 | 2143 | 39,623 |
| | 8 | S08 | 4046 | 2516 | 501 | 7063 |
| | 9 | S10 | 2981 | 1621 | 2392 | 6994 |
| | | **Physical** | | | | |
| | 10 | P01 | 47,031 | 3020 | 471 | 50,522 |
| | 11 | P03 | 4234 | 1251 | 142 | 5627 |
| Sydney, Australia | 12 | P04 | 25,123 | 3203 | 1025 | 29,351 |
| | 13 | P05 | 4220 | 6142 | 2051 | 12,413 |
| | 14 | P06 | 26,410 | 5171 | 521 | 32,102 |
| | 15 | P07 | 48,921 | 2422 | 718 | 52,061 |
| | 16 | P09 | 2722 | 1421 | 1022 | 5165 |
| | 17 | P10 | 11,102 | 1403 | 1501 | 14,006 |
| | | **Economic** | | | | |
| | 18 | E01 | 43,484 | 1206 | 2312 | 47,002 |
| | 19 | E03 | 2961 | 1341 | 724 | 5026 |
| | 20 | E04 | 39,991 | 2871 | 1011 | 43,873 |
| | 21 | E05 | 8420 | 4350 | 3251 | 16,021 |
| | 22 | E06 | 3771 | 4520 | 1861 | 10,152 |
| | **Total** | | **395,423** | **70,214** | **31,836** | **498,473** |
| | | **Cultural** | | | | |
| | 1 | C02 | 1521 | 745 | 254 | 2520 |
| | 2 | C05 | 2011 | 471 | 510 | 2992 |
| | | **Social** | | | | |
| | 3 | S02 | 3124 | 859 | 161 | 4144 |
| | 4 | S03 | 2861 | 1050 | 341 | 4252 |
| Aguita de la Perdiz, Chile | 5 | S07 | 3405 | 1015 | 172 | 4592 |
| | | **Physical** | | | | |
| | 6 | P01 | 5412 | 1251 | 108 | 6771 |
| | 7 | P03 | 4439 | 721 | 91 | 5251 |
| | 8 | P04 | 5039 | 681 | 211 | 5931 |
| | 9 | P05 | 1424 | 742 | 55 | 2221 |
| | 10 | P07 | 5697 | 1462 | 61 | 7220 |

**Table A1.** *Cont.*

| Case Study | S/N | Topics | negTweets | neuTweets | posTweets | ∑Tweets |
|---|---|---|---|---|---|---|
| | 11 | P11 | 1265 | 564 | 113 | 1942 |
| | 12 | P12 | 1123 | 558 | 333 | 2014 |
| | | Economic | | | | |
| | 13 | E01 | 4571 | 424 | 37 | 5032 |
| | 14 | E04 | 3961 | 751 | 91 | 4803 |
| | 15 | E05 | 516 | 224 | 961 | 1701 |
| | 16 | E07 | 1003 | 350 | 88 | 1441 |
| | 17 | E09 | 571 | 335 | 111 | 1017 |
| **Total** | | | **47,943** | **12,203** | **3698** | **63,844** |

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
