# Peer review of "Novel Use of Social Media Big Data and Artificial Intelligence for Community Resilience Assessment (CRA) in University Towns"

_sustainability, doi:10.3390/su15021295_

Round 1

Reviewer 1 Report

1(1) The authors provided scanty information about their online survey to validate the study text mining results ( see pages 8,16). Please provide a reference to the original source from which the questionnaires were obtained. Please describe the various stages of the development of the measuring instrument, including the pre-testing processes and the questionnaire administration period among others

2(2)  It is also unusual to report overall (all questions) Cronbach's alpha(CA) values without also reporting CA values for subscales (social, cultural, physical, economic and governance respectively). How was the overall CA of 0.879 and 0.803 obtained( see page 8, 16)? Does it represent the average CA scores across all five dimensions/subscales? The overall CA needs to be supported by previous studies.

3(3) The parameters of the online survey's reliability and validity need to be reported. Provide CA values for each scale and subscale, as well as factor loadings for those scales and subscales.

4(4) Figures 1 and 2 were appended to the manuscript without any corresponding textual description.

5(5)  The results were not empirically discussed (see section 5). Simply interpreting figures and tables does not constitute empirical discussion.

6(6)Managerial implication was scanty and too general( see page 21)

7(7) Tables 2, 3, and 5 would be much easier to read if the margins and text were aligned. The landscape orientation is an option.

8(8)The study orate that there are 10 ten community challenges but partial list were provides( see page 19 paragraph 3 ). List all the top 10 community challenges

9(9) Hong Kong should be included in the list of countries without any problems in their institutions or governments ( see page 19)

1(10)  Additionally, on page 19, paragraph one, indicate that S12 was a community-wide challenge only present in Hong Kong.

1(11) Loughborough's percentages for community problems were given incorrectly ( see page 19 paragraph 2)

Author Response

Dear Reviewer 1,

Thank you for agreeing to review our manuscript and giving us constructive feedback on improving it. We appreciate every comment you gave below and the precious time that you spent reading our work.

Kindly see our responses below to all 11 points raised. We hope that they meet your expectations and the standard of the Sustainability journal.

Kind regards,

The Authors.

Point 1: The authors provided scanty information about their online survey to validate the study text mining results ( see pages 8,16). Please provide a reference to the original source from which the questionnaires were obtained. Please describe the various stages of the development of the measuring instrument, including the pre-testing processes and the questionnaire administration period among others.

Response 1: Thank you for this great comment. Page 9 of the manuscript has been updated to contain the link to the Hong Kong Polytechnic University’s PhD thesis repository, where readers can find the questionnaire. The development and pre-testing phase of the questionnaire was also explained, and the reference to the study we followed was added for further reading. The 5-point scale was also explained. The survey took us seven months (from June 2020 to February 2021) due to the COVID-19 pandemic situation. This information was also added. Kindly see lines 303 – 326 on pages 8 and 9. Thank you.

Point 2: It is also unusual to report overall (all questions) Cronbach's alpha(CA) values without also reporting CA values for subscales (social, cultural, physical, economic and governance respectively). How was the overall CA of 0.879 and 0.803 obtained( see page 8, 16)? Does it represent the average CA scores across all five dimensions/subscales? The overall CA needs to be supported by previous studies.

Response 2: Thank you for bringing this error to our attention. We used a Python Panda for the analysis, and it gave both the CA for the subscales and the average CA (of the subclasses). Hence, we initially only reported the average of all the subscales (0.879) and also those only from the 6 case studies (0.803). We have corrected this error in the manuscript by putting the values for the subscales instead. Kindly see line 330 (page 9) and lines 403 - 406 (page 17). The methodology (Howard, 2022) we adopted for the Python CA calculation was also cited. Thank you so much for pointing this out.

Point 3: The parameters of the online survey's reliability and validity need to be reported. Provide CA values for each scale and subscale, as well as factor loadings for those scales and subscales.

Response 3: Thank you for re-emphasis this error. The corrections have been made on manuscript line 330 (page 9) and lines 403 - 406 (page 17). Kindly also find the codes and packages (Pingouin and Panda) we used for the Python CA analysis in this link: https://mattchoward.com/calculating-cronbachs-alpha-in-python/ This methodology is also cited in the updated manuscript.

Point 4: Figures 1 and 2 were appended to the manuscript without any corresponding textual description.

Response 4: Thank you for this comment. Textual descriptions have been appended to the manuscript on line 84 (page 2) and line 202 – 207 (page 5) for figures 1 and 2, respectively.

Point 5: The results were not empirically discussed (see section 5). Simply interpreting figures and tables does not constitute empirical discussion.

Response 5: Thank you for your comment. Section 5 has been strengthened to provide more discussion on the implications of the research outcomes. Kindly see lines 580 – 486 and 493 – 495 on page 22 and also section 5.2 on page 22.

Point 6: Managerial implication was scanty and too general( see page 21)

Response 6: We appreciate this comment. The managerial implications are discussed better in section 5 of the update manuscript.

Point 7: Tables 2, 3, and 5 would be much easier to read if the margins and text were aligned. The landscape orientation is an option.

Response 7: Sorry for any inconvenience caused while reading the tables. We totally agree with this comment. Tables 2, 3, and 5 were originally uploaded in landscape orientation, but the journal website converted all the pages to portraits in the Microsoft word version of the manuscript. If you can download the uploaded PDF version instead of the word version, we believe the tables would look better.

Point 8: The study orate that there are 10 ten community challenges but partial list were provides( see page 19 paragraph 3 ). List all the top 10 community challenges

Response 8: Thank you for this comment. The list has been updated. Kindly see lines 467 - 479 on pages 21 and 22.

Point 9: Hong Kong should be included in the list of countries without any problems in their institutions or governments ( see page 19)

Response 9: We appreciate this comment but cannot find the exact correction you referred to on page 19. We are unsure if you were referring to the “governance and institutional community challenges” in Hung Hom (Hong Kong) due to studentification. It is worth noting that Hong Kong is officially a Special Administrative Region (SAR) of the People's Republic of China and not a country. Hung Hom is a district in Hong Kong SAR, China.

Point 10: Additionally, on page 19, paragraph one, indicate that S12 was a community-wide challenge only present in Hong Kong.

Response 10: Thank you for pointing out this omission. It has been corrected in lines 438 – 439 of the manuscript.

Point 11: Loughborough's percentages for community problems were given incorrectly ( see page 19 paragraph 2)

Response 11: Thank you so much for identifying this error. The percentage of economic challenges in Loughborough has been corrected on line 452.

Reviewer 2 Report

The authors explored how UGC from microblogs can identify community challenges using artificial intelligence (ML and NLP) tools like LDA and VADER. The study shows that all 45 challenges clustered around the five community resilience dimensions were accepted as negative impacts of studentification.  The research also contributes to knowledge of research in the new normal by proving that longitudinal studies can be done remotely. The overall rating of the manuscript is good. The manuscript is recommended for publication in Sustainability.

Author Response

The authors explored how UGC from microblogs can identify community challenges using artificial intelligence (ML and NLP) tools like LDA and VADER. The study shows that all 45 challenges clustered around the five community resilience dimensions were accepted as negative impacts of studentification.  The research also contributes to knowledge of research in the new normal by proving that longitudinal studies can be done remotely. The overall rating of the manuscript is good. The manuscript is recommended for publication in Sustainability.

Response:

Dear Reviewer 2,

Thank you for agreeing to review our manuscript and recommending it for publication. We are grateful.

Kind regards,

The Authors.

Reviewer 3 Report

The reviewers raised the following questions, and the authors are invited to carefully consider and revise:

Author Response

Reviewer: “Novel use of social media big data and artificial intelligence for Community Resilience Assessment (CRA) in university towns: Sustainability-2057624. First, The Reviewer would like to thank the editors of "Sustainability" for sending me the manuscript for review. The reviewers also cherish this opportunity to review the manuscript and will carefully review the manuscript.

The research process of the manuscript is complete. However, the significant problems are the establishment of the research model, the analysis of the data of the research process, and the disclosure of the research innovation results need to be more precise, prominent, bright, and robust. The research process uses a large amount of data accumulation so that the research value and research significance of the manuscript are only partially explored and reflected. There are many errors in the English expression of the manuscript, and the full text needs to be proofread.

The final review result of the manuscript is: The reviewer recommends that the article be Major revision; the current format is not suitable for publishing this manuscript.

Response:

Dear Reviewer 3,

Thank you for agreeing to review our manuscript and giving us constructive feedback on improving it. We appreciate every comment you gave below and the precious time that you spent reading our work.

Kindly see our responses below to all 9 points raised. We hope that they meet your expectations and the standard of the Sustainability journal.

Kind regards,

The Authors.

The reviewers raised the following questions, and the authors are invited to carefully consider and revise:

Point 1: Page 1; The format of the keywords is wrong, control 6-7, and delete all duplicates with the title.

Response 1: Thank you for this comment. The duplicates with the title have been removed.

Point 2: Page 1-2; In the Introduction, the author's research questions, and literature analysis are not particularly clear, and the innovation aspect of the research is briefly mentioned. These two aspects need to be supplemented and clarified.

Response 2: We appreciate you pointing this out. The corrections have been made. Kindly see lines 67 – 79 (page 2) of the updated manuscript.

Point 3: Page 3-6; Section 2 of the manuscript explains several conceptual terms and does an extensive literature survey. The theoretical model is established in the last two paragraphs of the text. The problem is that part of the literature survey can be placed directly in the first section. After selecting the latest literature and passing the survey, directly explain the research background and existing challenges, and introduce in detail the difference between theoretical model building and published literature models, and your model is the better. What problem was solved? This section needs significant revision and improvement.

Response 3: Thank you for this comment. We believe new changes made to section 1 of the updated manuscript already took care of most of the issues highlighted here. The updated section 1 explains what problem was solved and the novelty of the new framework. Section 2 was also updated to reflect on the “problem solved” and the advantages the proposed framework has over the existing ones. We maintained section 2 instead of collasping it into section 1 so that section 1 wont be too long.

Point 4: Page 7; The font of Figure 3 needs to be modified, check the full text.

Response 4: Thank you so much for pointing out this error. The font size of the Palatino Linotype text changed when the figure was converted to a picture. We have reduced it to the right font size in the updated manuscript.

Point 5: Page 9-19; The table needs to be formatted, and the current data is messy, which affects reading. The manuscript lists many calculation data, lacking the scientific and validity proof of the critical data description and the application research model framework to analyze the data.

Response 5: Thank you for this comment. Tables 2, 3, and 5 were designed in landscape orientation but the journal website changes all the pages in the manuscript to portrait orinetation, hence the messy data. A PDF format of the updated manuscript has been uploaded during this resubmission, we hope you can access that at your end to read the tables better. We are sorry for the inconvenience caused in reading those tables in the word format of the first version of the manuscript.

We used Python programmes for most of the big data analysis (machine learning and natural language processing). The codes we modified for our proposed framework were included in the appendix. Links to the liberies used were also appended as footnotes. The framework we adopted and improved upon in this study was also published in Cities (https://doi.org/10.1016/j.cities.2020.102986). We did not see the need to include all the models, algorithms, and programming codes that are recently developed and well documented in the literature, so we cited them in the manuscript instead. Besides, that will make the article very long and messy (full of programming codes). We hope, this explanation justifies the problem you identified with the research data analysis. We appreciate you pointing this out and we are willing to make improvement, if need be. Thank you.

Point 6: Page19; The discussion part of the fifth section requires the author to make a comprehensive and detailed analysis of the analyzed data, especially the innovative conclusions that should be clearly explained. This section needs further strengthening.

Response 6: Thank you for this comment. Section 5 of the manuscript has been strengthened accordingly. Kindly see lines 438 – 439, 467 – 486, and 488 – 512 on pages 21 – 22. Thank you for this recommendation.

Point 7: Page 1-20; Please add the units of Figure abscissa and ordinate and check the full text.

Response 7: Dear reviewer, thank you for your comments. We are not sure if we understand this particular comment properly. Pages 1 – 20 of the manuscript do not contain any 2-dimensional graph with x and y coordinates that may require units.

Point 8: Page 28-31; Please revise and check the wrong format of the citation; some citations cannot be retrieved.

Response 8: Thank you for calling our attention to this. The citations were update using EndNote and citation style downloaded from https://www.mdpi.com/authors/references

Point 9: Page 1-21; Please check the entire manuscript in English for wording, spelling, and multiple misuses of long sentences. The manuscript must be proofread by a professional English professor from a native-speaking country before it can be submitted.

Response 9: Thank you for this comment. The manuscript has been proofread as advised.

Reviewer 4 Report

This manuscript adopted artificial intelligence method including Text Mining, Machine Learning and Natural Language Processing to carry out community resilience assessments from cultural, social, physical, economic, and institutional and governance community challenges based on social media big data. This research has a certain practical significance for the governance and planning of the university town. However, the manuscript still has the following problems.

1)    The process of processing microblog data using artificial intelligence models such as Text Mining, Machine Learning and Natural Language Processing needs to be described in detail.

2)    It is suggested to design a comprehensive index model for the resilience of university towns, calculate the scores of six university towns, and propose planning and management suggestions according to the evaluation results.

3)    Some sentences are too long to understand.

Author Response

This manuscript adopted artificial intelligence method including Text Mining, Machine Learning and Natural Language Processing to carry out community resilience assessments from cultural, social, physical, economic, and institutional and governance community challenges based on social media big data. This research has a certain practical significance for the governance and planning of the university town. However, the manuscript still has the following problems.

Dear Reviewer 4,

Thank you for agreeing to review our manuscript and giving us constructive feedback on improving it. We appreciate every comment you gave below and the precious time that you spent reading our work.

Kindly see our responses below to the points raised. We hope that they meet your expectations and the standard of the Sustainability journal.

Kind regards,

The Authors.

1)    The process of processing microblog data using artificial intelligence models such as Text Mining, Machine Learning and Natural Language Processing needs to be described in detail.

Response: Thank you very much for your comment. This manuscript is part of a bigger research that has 3 parts. The first part developed models and algorithms used for mining and pre-processing textual big data from microblogs. The mining and pre-processing framework was tested using a case study and was published in Cities.

This manuscript is the second part of the research. In this study, we adopted the first mining and pre-processing framework, improved its data validation phase, and used the algorithms to develop a community resilience assessment framework. We used it in 6 case studies experiencing studentification and we were able to demonstrate that it can be used for community resilience assessment. The new framework was also able to identify community challenges in the 5 dimensions of resilience and provide spatiotemporal analysis. These were two research gaps identified in the community resilience and studentification literature.

The third part of the research is a manuscript currently under review in Sustainability. It develops a localized community resilience index using the AI-based community resilience assessment framework developed in this manuscript. One of the 6 case studies (Akoka in Lagos, Nigeria) was used as a case study to demonstrate the process of developing a localized community resilience index for university towns. Delphi and AHP were used to develop the index. The index has both outcome indicators and process indicators (planning and management suggestions).

So, instead of writing the processes again in details in this manuscript, we cited the first paper (https://doi.org/10.1016/j.cities.2020.102986) and gave a summary of the mining and preprocessing aspects and paid more attention to the parts we modified and improved. We also added the codes we modified to the appendix of this manuscript.

This manuscript and the one for the index were one manuscript before but it was too long, so we were asked to slit in into two manuscripts.

2)    It is suggested to design a comprehensive index model for the resilience of university towns, calculate the scores of six university towns, and propose planning and management suggestions according to the evaluation results.

Response: We are glad you mentioned this. Kindly see our comments above. We did the exact thing,  but to keep the manuscript within a reasonable length, we had to split it into two manuscripts.

3)    Some sentences are too long to understand.

Response: The manuscript has been proofread and the language improved. Thank you for pointing this out.

Reviewer 5 Report

This paper studies community challenges by analyzing Twitter data using some open-source machine learning libraries. Overall it's an interesting application of social media data. I have a few questions, though:

  1. The entire analysis is based on the assumption that the data collected is comprehensive enough. However, the tool used for scraping tweets claims that "GetOldTweets-Python is a project written in Python to mine old and backdated tweets, It bypasses **some** limitations/restrictions of the Twitter API." Do the authors have some understanding of the data completeness? If the data is highly skewed, that would significantly affect the analysis result.
  2. Are there any duplicated tweets scraped? 
  3. Tables 2, 3 and 5 are unreable, with text and numbers being cut into multiple lines. Plus, they provide way too much information to readers. Please consider removing them or moving them into the appendix.
  4. The paper needs to be carefully proofread. There are typos here and there. To name a few. In section 2.3, it was said, "This has made User-Generated Contents (UGU)." I suppose it's UGC. In section 6, it was said that "First, a programmatic algorithm was used to **mind** the big data." Should it be mine? 
  5. Minor: in section 2.3, the authors mention that **billions of data** are generated in cities. Data alone is uncountable. Do you mean data points? Could you be more specific? 

Author Response

This paper studies community challenges by analyzing Twitter data using some open-source machine learning libraries. Overall it's an interesting application of social media data. I have a few questions, though:

Dear Reviewer 5,

Thank you for agreeing to review our manuscript and giving us constructive feedback on improving it. We appreciate every comment you gave below and the precious time that you spent reading our work.

Kindly see our responses below to the points raised. We hope that they meet your expectations and the standard of the Sustainability journal.

Kind regards,

The Authors.

The entire analysis is based on the assumption that the data collected is comprehensive enough. However, the tool used for scraping tweets claims that "GetOldTweets-Python is a project written in Python to mine old and backdated tweets, It bypasses **some** limitations/restrictions of the Twitter API." Do the authors have some understanding of the data completeness? If the data is highly skewed, that would significantly affect the analysis result.

Response: Thank you for this comment. We have modified the original “GetOldTweets-Python project” and improved the algorithm. The new programmatic algorithm “Optimized-Modified-GetOldTweets3-OMGOT” streamlines searches better and by-passes the rate limits of the Twitter APIs, allowing the download of unlimited historic tweets generated from a specific geo-location using the PyQuery tool, from terminal or command prompt. Due to privacy, Twitter made it harder to access the exact geolocation of tweets within 3sqm. For example, setting the tool to download tweets from an office desk or a bedroom only instead of the whole office floor/block or the whole house and beyond. This is to prevent spying/trolling. The original library (GetOldTweets-Python) left that limitation. We also decided to respect that privacy rule. Therefore, both tools do not bypass ALL the Twitter API restrictions. This 1 limitation does not affect the comprehensiveness of our data because we are minig data from communities and towns that are bigger than 3 square metres.

The mining and pre-processing framework is made up of 3 key tools: Optimized-Modified-GetOldTweets3-OMGOT, Latent Dirichlet Allocation, and Valence Aware Dictionary and sEntiment Reasoner. We first published it in Cities journal and now we decided to modify it again and use it for community resilience assessment in university towns.

Are there any duplicated tweets scraped?

Response: Thank you for this comment. The tools does not scrap duplicates, except there is a modification. However, the meta data will show how many times a tweet has been retweeted, liked, or commented on, etc and by who (username).

Tables 2, 3 and 5 are unreable, with text and numbers being cut into multiple lines. Plus, they provide way too much information to readers. Please consider removing them or moving them into the appendix.

Response: Thank you for pointing this out. Tables 2, 3, and 5 were designed in landscape orientation but the journal website changes all the pages in the manuscript to portrait orinetation, hence the messy data. A PDF format of the updated manuscript has been uploaded during this resubmission, we hope you can access that at your end to read the tables better. We are sorry for the inconvenience caused in reading those tables in the word format of the first version of the manuscript.

The paper needs to be carefully proofread. There are typos here and there. To name a few. In section 2.3, it was said, "This has made User-Generated Contents (UGU)." I suppose it's UGC. In section 6, it was said that "First, a programmatic algorithm was used to **mind** the big data." Should it be mine?

Minor: in section 2.3, the authors mention that **billions of data** are generated in cities. Data alone is uncountable. Do you mean data points? Could you be more specific?

Response: Thank you for this comment. We agree with your comment on “billions of data”. It was meant to be “data points”. This has been corrected in the updated manuscript. We have also tried to correct the typos and proofread the manuscript. We really appreciate this comments.

Round 2

Reviewer 3 Report

The reviewers raised the following questions, and the authors are invited to carefully consider and revise:

1.     Page 3-6; It is suggested that the author reduce 2. The number of theoretical and conceptual background references has been cited 61 [20-80]. Is it necessary to quote so many?

2. Page 7; Figure 3 and Figure in the literature published by the author. It's almost the same. Please edit it for review. Is it in line with the publishing regulations (https://doi.org/10.1016/j.cities.2020.102986)?

Author Response

Dear Reviewer 3,

Compliment of the season!

Thank you for agreeing to review our manuscript for the second time and giving us more constructive feedback to make it better. We appreciate every comment you gave below and the precious time that you spent reading our work, again.

Kindly see our responses below to the 2 points raised.

Thank you once again.

Kind regards,

The Authors.

Point 1: Page 3-6; It is suggested that the author reduce 2. The number of theoretical and conceptual background references has been cited 61 [20-80]. Is it necessary to quote so many?

Response: Thank you for this comment. Section 2 has been reduced. Paragraphed 2 of section 2.1 has been removed. The references in the whole section 2 (pages 3-6) have been reduced from 61 to 37 [20-56]. We really appreciate this feedback.

Point 2: Page 7; Figure 3 and Figure in the literature published by the author. It's almost the same. Please edit it for review. Is it in line with the publishing regulations (https://doi.org/10.1016/j.cities.2020.102986)?

Response: Thank you for pointing this out. Figure 3 on page 7 has been further modified, and the steps were also edited in line with the new figure. We have re-checked to ensure that there are no copyright infringements with https://doi.org/10.1016/j.cities.2020.102986 since it was cited in the title of the figure. We honestly appreciate your feedback on this.

Reviewer 4 Report

The manuscript still has the following problems.

1)    The process of processing microblog data using artificial intelligence models such as Text Mining, Machine Learning and Natural Language Processing needs to be described in detail.

2)    It is suggested to design a comprehensive index model for the resilience of university towns, calculate the scores of six university towns, and propose planning and management suggestions according to the evaluation results.

3)    Some sentences are too long to understand.

Author Response

Dear Reviewer 4,

Compliment of the season!

Thank you for agreeing to review our manuscript for the second time. We have received your comments from the second round of reviews. However, we have observed that the comments were exactly the same from the first round. We are unsure if you have seen our responses to the same comments in the first review round. So, kindly find attached our responses again for your kind perusal.

We look forward to your feedback and more constructive comments to improve the quality of our manuscript.

Thank you.

Kind regards,

The Authors.

Comments and responses

1) The process of processing microblog data using artificial intelligence models such as Text Mining, Machine Learning and Natural Language Processing needs to be described in detail.

Response: Thank you very much for your comment. This manuscript is part of a bigger research that has 3 parts. The first part developed models and algorithms used for mining and pre-processing textual big data from microblogs. The mining and pre-processing framework was tested using a case study and was published in Cities.

This manuscript under review is the second part of the research. In this study, we adopted the first mining and pre-processing framework, improved its data validation phase, and used the algorithms to develop a community resilience assessment framework. We used it in 6 case studies experiencing studentification, and we were able to demonstrate that it can be used for community resilience assessment. The new framework was able to identify community challenges in the 5 dimensions of resilience and provide spatiotemporal analysis. These were research gaps identified in the community resilience and studentification literature.

The third part of the research is a manuscript currently under review in Sustainability. It develops a localized community resilience index using the AI-based community resilience assessment framework developed in this manuscript. One of the 6 case studies (Akoka in Lagos, Nigeria) was used as a case study to demonstrate the process of developing a localized community resilience index for university towns. Delphi and AHP were used to develop the index. The index has both outcome indicators and process indicators (planning and management suggestions).

So, instead of writing the processes and codes again in detail in this manuscript, we cited the first paper (https://doi.org/10.1016/j.cities.2020.102986) and gave a summary of the mining and preprocessing aspects and paid more attention to the parts we modified and improved. We also added the codes we modified to the appendix of this manuscript.

This manuscript and the one for the index were one manuscript before, but it was too long, so we were asked to slit in into two manuscripts.

2)    It is suggested to design a comprehensive index model for the resilience of university towns, calculate the scores of six university towns, and propose planning and management suggestions according to the evaluation results.

Response: We are glad you mentioned this. Kindly see our comments above. We did the exact thing.  To keep the manuscript within a reasonable length, we had to split it into two manuscripts: This and the other titled: A Composite Resilience Index (CRI) for Developing Resilience and Sustainability in University Towns (Manuscript ID: sustainability-2087430) under review in this special issue of sustainability.

3)    Some sentences are too long to understand.

Response: The manuscript has been proofread, and the language improved. Thank you for pointing this out.

Reviewer 5 Report

The authors addressed my feedback. Great work!

Author Response

Reviewer: The authors addressed my feedback. Great work!

Response:

Dear Reviewer 5,

Compliment of the season!

Thank you for agreeing to review our manuscript for the second time and recommending it for publication. We are happy that our responses adequately addressed your constructive feedback and improved the quality of the manuscript. We are grateful for your time and effort during this festive period. Thank you.

Kind regards,

The Authors.

Round 3

Reviewer 4 Report

The manuscript has been well modified as required, please accept!